# Challenges and opportunities of the full phase-out of fossil fuels under the 1.5 °C goal

Shotaro Mori [1,2] ✉, Siddharth Joshi [2], Volker Krey [2], Ken Oshiro [3], Oliver Fricko [2], Takuya Hara [2] & Shinichiro Fujimori [1,2,4] ✉

The COP28 decision called for transitioning away from fossil fuels, sparking a growing interest in their full phase-out. However, energy system transformation pathways towards a phase-out of fossil fuels, which may reduce the reliance on carbon dioxide removal to meet the 1.5 °C goal, remain unclear. Here, we employ two global energy system models to explore energy system transformations and the challenges and opportunities associated with attaining a full phase-out of fossil fuels. We found that phasing out fossil fuels by 2050 would require accelerating direct and indirect electrification, involving 1.6–1.8-fold increases in power generation compared to the conventional cost-effective 1.5 °C pathways. This transition from cost-effective to fossil fuel phase-out pathways would increase energy supply investments by up to 34% over this century and require accelerated deployment of solar and wind power, as well as electrolysers. Despite opportunities including lower reliance on carbon dioxide removal and increasing probability of returning to 1.5 °C after temperature overshoot, these additional requirements imply that international society must approach the transition towards zero-fossil energy systems with strong determination.

Limiting global warming to well below 2 °C, and pursuing efforts to return to 1.5 °C after temporary overshoot by the end of the century in line with the Paris Agreement, requires a rapid reduction of fossil fuel contributions to the energy system[1,2]. The final decision of the first Global Stocktake (GST-1) at the 2023 United Nations Conference of the Parties (COP28) highlighted the necessity for urgent action to address the climate crisis, given that the Parties are not on track to achieve the long-term goals of the Paris Agreement[3]. A key achievement of the GST-1 decision was that it called for all Parties to contribute to global efforts, particularly transitioning away from fossil fuels in energy systems. Although the final decision does not explicitly mention a full phase-out, momentum is growing for the phase-out of fossil fuels to become a key focus of climate policies moving forward from COP28. Notably, in 2024, the Group of Seven (G7) committed to phasing out existing unabated coal power generation during the first half of the 2030s[4], and the Group of Twenty (G20) leaders fully subscribed to the outcome of COP28, in particular the UAE Consensus and the GST-1[5].

The decarbonisation and defossilisation of energy systems could follow different energy transformation pathways. These differences may concern the pace, sequence, and stringency of fossil fuel phase-out, especially in sectors that are technically and economically challenging to decarbonise. Scenarios from integrated assessment models (IAMs) and energy system models have provided insights into energy transformation pathways in line with the long-term goals of the Paris Agreement[6–12], as reflected in assessment reports by the Intergovernmental Panel on Climate Change (IPCC), e.g., the Sixth Assessment Report (IPCC AR6)[13]. To obtain robust insights from previous studies, thousands of scenarios were submitted to the IPCC AR6 scenario database for analysis by IPCC Working Group III[14]. These scenario ensembles suggested that cost-effective decarbonisation pathways involve replacing fossil fuel consumption primarily through direct electrification with renewable electricity, while also addressing $CO_2$ emissions from hard-to-abate fossil fuel consumption through incorporating carbon

[1]Kyoto University, Kyoto, Japan. [2]International Institute for Applied System Analysis (IIASA), Laxenburg, Austria. [3]Hokkaido University, Sapporo, Japan. [4]National Institute for Environmental Studies (NIES), Tsukuba, Japan. ✉e-mail: mori.shotaro.2n@kyoto-u.ac.jp; fujimori.shinichiro.8a@kyoto-u.ac.jp

capture and storage (CCS) and offsetting via carbon dioxide removal (CDR)[15].

However, achieving what could be called a zero-fossil (ZF) energy system, beyond transitioning away from and phasing out fossil fuels, will likely require pathways that depart from these typical decarbonisation pathways. Such differences are likely to arise because ZF energy systems require the elimination of even hard-to-abate fossil fuel consumption, which would remain in a typical 1.5 °C energy system[16–19]. Based on the outcomes of previous studies that focused on reducing residual fossil $CO_2$ emissions considering sustainability concerns about large-scale CDR deployment[20,21], ZF energy systems are expected to rely on various measures such as deep electrification[22,23], the utilisation of alternative fuels like biofuels[24,25] and hydrogen-based energy carriers (e.g., hydrogen itself, ammonia, and synthetic hydrocarbon fuels)[19,26], increasing energy efficiency in end-use sectors, and lowering energy service demand[27,28]. Considering the features of each measure, such as the impacts of large-scale bioenergy use on food security and biodiversity[20,29–31] and increased mitigation costs associated with the extensive use of hydrogen-based energy carriers[32,33], determining how to integrate these measures to achieve ZF energy systems remains a critical question.

Previous studies have limited insight into the pathways for reaching ZF energy systems. In the context of exploring cost-effective decarbonisation pathways aligned with the 2 °C and 1.5 °C goals of the Paris Agreement, there has been no explicit need to create ZF energy systems. A more technical factor is the lack of models that incorporate technology options for the full phase-out of residual fossil fuel consumption on the end-use side. Indeed, scenarios within the IPCC AR6 scenario database, which limit temperature increases to 1.5 °C with 50% likelihood in 2100 (categories C1 and C2; see Methods), show primary energy supplies from fossil fuels of 115–334 EJ/yr in 2050 and 30–287 EJ/yr in 2100 (10th to 90th percentiles)[14]. Furthermore, there are no AR6 scenarios that achieve the full phase-out of fossil fuels at any point within this century[25]. Although some recent studies focusing on 100% renewable energy systems can be interpreted as a subset of ZF scenarios[34,35], they have typically assumed a constrained set of available technologies (e.g., excluding nuclear energy or CDR), focused on limited numbers of sectors (e.g., power or energy supply), or relied on prescribed transformation pathways for energy end uses[22,36–38]. After COP28, societies may increase their focus on phasing out fossil fuels, which historically have been the main drivers of climate change. Furthermore, growing interest in hydrogen-based energy carriers and carbon capture, utilisation, and storage (CCUS) as measures to address residual fossil $CO_2$ emissions[39,40] has led to the emergence of models that include these potential enablers of ZF energy systems[19,23,26].

In this study, we investigated the extent to which the full phase-out of fossil fuels would differ from typical 1.5 °C scenarios and identified the challenges and opportunities associated with their realisation. To explore two distinct illustrative pathways for ZF energy systems and gain shared insights from the model ensemble, we employed two global energy system models: AIM-Technology (Asia–Pacific Integrated Model-Technology, hereinafter AIM)[12], and MESSAGEix-GLOBIOM (Model for Energy Supply Strategy Alternatives and their General Environmental Impact combined with the Global Biosphere Management Model, hereinafter MESSAGEix)[41–43]. We defined the entire energy sector, including non-energy use, as the boundary of the energy system, and ZF was defined as the full phase-out of coal, oil, and natural gas. A scenario-based approach was adopted to understand the diverse transition pathways to ZF energy systems, characterised primarily by the target year for achieving ZF. The ZF scenarios were labelled according to target years, from 2050 (ZF2050) to 2100 (ZF2100) in 10-year increments. These scenarios imposed upper limits on the primary energy supply from fossil fuels, along with emission constraints corresponding to a carbon budget of 500 $GtCO_2$ from 2018 to 2100, covering $CO_2$ emissions from all sectors[10]. Finally, we ran a model-specific 1.5 °C scenario (Opt1.5 C) that imposes only emission constraints, without setting limits on the primary energy supply from fossil fuels. The results were compared with those for the ZF scenarios, and the additional efforts required to achieve ZF energy systems were examined. Additionally, we used the C1 and C2 scenarios obtained from the AR6 scenario database[14] as examples of typical 1.5 °C energy systems for comparison with the ZF scenarios.

## Results

### Fossil fuel phase-out in energy systems

The primary energy mix of the ZF energy system in 2050 exhibited unique characteristics compared to those of the AR6 C1 and C2 scenarios (Fig. 1a). In 2050, fossil fuels accounted for 35% (188 EJ/yr) of the total primary energy supply according to AIM, and for 54% (275 EJ/yr) according to MESSAGEix, for the Opt1.5 C scenarios (Fig. 1a, d). As energy systems approached ZF, the primary energy mix shifted away from the distribution of typical 1.5 °C scenarios, including the Opt1.5 C and AR6 C1 and C2 scenarios, due to the increasing replacement of fossil fuels with non-fossil energy, particularly non-biomass renewables. By contrast, in 2100, as deeper decarbonisation occurred, the primary energy mix of ZF scenarios closely resembled those of the Opt1.5 C scenarios, highlighting the difficulty of achieving near- to midterm transition to ZF energy systems (Supplementary Fig 1a). The cumulative fossil fuel primary energy supply from 2020 to 2100 was reduced to 34–43% of the Opt1.5 C scenarios in the ZF2050 scenarios and to 64–70% even in the ZF2100 scenarios (Fig. 1e).

Compared to the unique primary energy mix of the ZF energy system, the share of individual energy sources in the power generation mix did not significantly deviate from that of the AR6 C1 and C2 scenarios (Fig. 1b), because fossil fuels in the power sector are phased out to an extent that is nearly equal to a full phase-out, as observed in most previous studies[44]. In the Opt1.5 C scenario of MESSAGEix, natural gas power plants with CCS contribute as both a flexible generator and a bridging technology, constituting the majority of fossil fuel with CCS in this scenario. However, even in the ZF2100 scenario, where the target year for ZF was set at the end of the century, the role of fossil power generation was limited by 2050 (Fig. 1f and Supplementary Fig. 2a). In 2100, the power generation mixes of ZF scenarios and the Opt1.5 C scenario were closely aligned within each model (Supplementary Fig. 1b).

The final energy mix of the ZF energy system was characterised by a lower share of liquid and gaseous fuels and a higher share of non-hydrocarbon fuels, which mainly consist of electricity and hydrogen, compared to those in the Opt1.5 C and the AR6 C1 and C2 scenarios (Fig. 1c). In the ZF2050 scenarios, the share of non-hydrocarbon fuels within total final energy consumption reached 67–80% by 2050. Notably, MESSAGEix exhibited a more pronounced transformation on the energy demand side compared to AIM. The main reason for this difference was the modelling of fuel switching on the end-use side, particularly in the scope of energy service demands that can be met by non-hydrocarbon technologies. In addition, MESSAGEix used in this study does not include direct air capture (DAC) and consequently does not consider DAC-based synthetic fuels, known as e-fuels, which further contributed to the difference. While the share of solid fuels decreased to 5–11% in the ZF2050 scenarios, it remained comparable to those in the AR6 C1 and C2 scenarios. Similar to the power generation mix, the final energy mix of the ZF scenarios and the Opt1.5 C scenario within each model was closely aligned by 2100 (Supplementary Fig. 1c).

### Energy demand transformation

Approaches to phasing out fossil fuels on the energy demand side showed some differences across models and scenarios (Fig. 2). In 2050, the Opt1.5 C scenarios decarbonised end-use sectors primarily

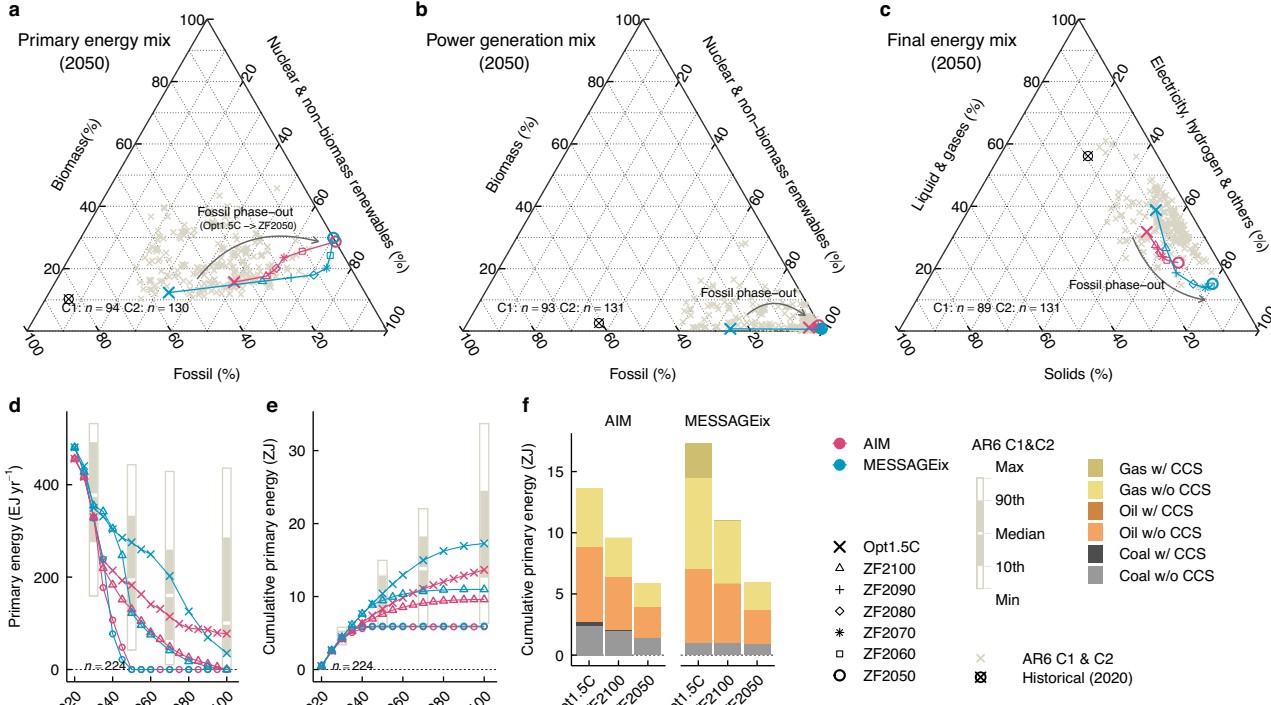

**Fig. 1 | Fossil fuel phase-out and energy system transformation.** Energy mix for primary energy (**a**), power generation (**b**), and final energy (**c**) in 2050 shown as ternary diagrams. In these diagrams, the share of each component should be read from ticks parallel to the edge where that component equals zero. Coloured symbols represent energy mixes by scenario and model. Grey symbols represent the energy mixes of the Intergovernmental Panel on Climate Change Sixth Assessment Report (IPCC AR6) C1 and C2 scenarios for 2050. Black symbols represent the historical energy mix for 2020, based on the International Energy Agency (IEA) energy balance[68]. Annual primary energy supply from fossil fuels from 2020 to 2100 (**d**) and cumulative primary energy supply from fossil fuels from 2020 to 2100 (**e**). Box plots illustrate the primary energy supply from fossil fuels in the IPCC AR6 C1 and C2 scenarios for 2030, 2050, 2070, and 2100. Annual primary energy supplies from coal, oil, and natural gas are shown in Supplementary Fig. 1d–f. **f** Cumulative primary energy supply from fossil fuels by fuel type from 2020 to 2100. "*n*" denotes the number of available scenarios in each category. In this study, primary energy accounting was based on the direct equivalent method, which systematically reduces the contribution of non-combustible energy sources such as hydro, nuclear, solar, and wind energy compared to combustible fuels[69].

with electricity, accounting for 39–47% of the total final consumption, supported by biomass and hydrogen, while fossil fuel consumption, mainly oil products and natural gas, remained at 34–39% (Fig. 2a, b). In the ZF2050 scenario of AIM, the full phase-out of fossil fuels in end-use sectors by 2050 was achieved by replacing residual fossil fuel consumption with the expanded use of biomass, hydrogen, and synthetic fuels in the industry and transport sectors (Supplementary Fig. 3b, c). By contrast, in the ZF2050 scenario of MESSAGEix, deeper direct electrification in the industry and buildings sectors, expanded biomass use in the industry and transport sectors, and increased hydrogen use in the transport sector contributed to achieving ZF in end-use sectors by 2050 (Supplementary Fig. 3b–d). By 2100, the share of fossil fuels in total final consumption had already decreased to 5–12%, even in the Opt1.5 C scenarios, such that the transition to ZF scenarios involved simply replacing this residual fossil fuel consumption with biomass or synthetic fuels (Fig. 2b).

The development of indirect electrification through hydrogen-based energy carriers and biomass utilisation to achieve the full phase-out of fossil fuels by the middle of this century was a strategy commonly seen in both models (Fig. 2b). The trajectory of hydrogen penetration was similar across models. The Opt1.5 C scenarios gradually reached hydrogen shares of 13–17% by the end of the century, while the ZF2050 scenarios achieved this level earlier, by the middle of the century. Synthetic fuels were a non-fossil option unique to AIM, accounting for 10% of the final consumption in ZF energy systems. By contrast, in the Opt1.5 C scenario, their share in total final consumption remained around 1%, even by the end of the century, indicating that their use expands only under the extreme conditions of ZF scenarios.

In the ZF2050 scenario of MESSAGEix, a sharp increase in direct electrification rates occurred to achieve the full phase-out of fossil fuels by 2050. By contrast, the increase in direct electrification rates from the Opt1.5 C scenario to the ZF2050 scenario was more modest in AIM compared to MESSAGEix, likely because AIM had already reached near-maximum levels of direct electrification in its Opt1.5 C scenario. As discussed later, in the ZF scenarios, almost all hydrogen was derived from electrolysis powered by renewable electricity (Supplementary Fig. 2b), reaffirming the effectiveness of hydrogen-based energy carriers in extending the application of renewables to sectors where direct electrification is challenging. The share of biomass in the total final consumption peaked at around 20% by the middle of the century, and then gradually declined to approximately 10% by 2100. This finding suggests that while biomass played a critical role in achieving ZF by the middle of the century, its contribution became smaller towards the end of the century due to growth in the total final consumption and biomass supply limitations.

The differences in target years with respect to achieving a full phase-out of fossil fuels significantly influenced the pace of transitions in end-use sectors (Fig. 2b). Achieving a full phase-out of fossil fuels by the middle of this century requires realising an energy transition in end-use sectors earlier than transitions that gradually occurred towards the end of the century in the Opt1.5 C scenarios. By contrast, pushing back the target year to the end of the century led to transitions in end-use sectors that more closely followed the pathways of the Opt1.5 C scenarios, and proceeded at a more gradual pace. In particular, the trajectory of biomass shares indicates that the mid-century peak observed in the ZF scenarios can be avoided, implying that a

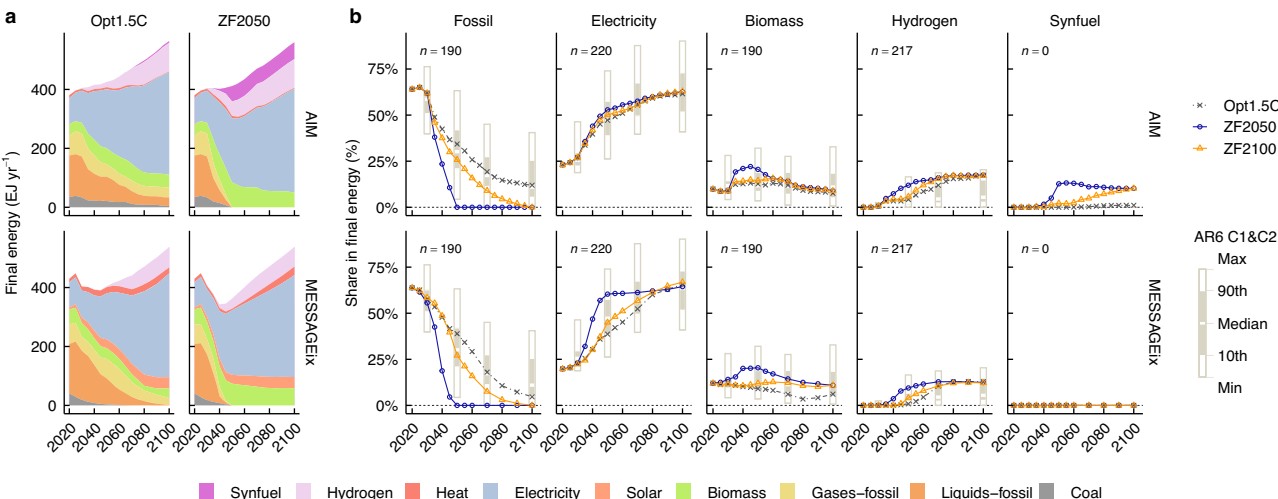

**Fig. 2 | Transformation to ZF energy systems in energy demand sectors. a** Final energy consumption in the Opt1.5 C and ZF2050 scenarios during the period 2020–2100. Results for other scenarios are shown in Supplementary Fig. 3a. **b** Shares in total final energy consumption in the Opt1.5 C, ZF2100, and ZF2050 scenarios during the period 2020–2100. Box plots illustrate the shares in final energy of the IPCC AR6 C1 and C2 scenarios for 2030, 2050, 2070, and 2100. "*n*" denotes the number of available scenarios in each category.

rapid scale-up in energy crop production can also be avoided (Supplementary Fig. 4).

## Energy supply transformation

Substantial increases in power and hydrogen generation by the middle of this century are among the most prominent characteristics of the ZF scenarios (Fig. 3). In 2100, total power generation in the ZF scenarios increased by approximately 10–20% compared to the Opt1.5 C scenario, but power generation levels across all scenarios in this study, including the ZF scenarios, were within the range of the AR6 C1 and C2 scenarios (Fig. 3c). However, in 2050, significant increases in total power generation, particularly from solar and wind energy, were observed in the ZF scenarios (Fig. 3a, b). Notably, in the ZF2050 scenario of AIM, total power generation in 2050 increased 1.6-fold compared to the Opt1.5 C scenario, reaching levels significantly exceeding the maximum observed in the AR6 C1 and C2 scenarios (Fig. 3c). Similarly, although the increase in absolute terms was not as pronounced, total power generation increased in the ZF2050 scenario of MESSAGEix, reaching 1.8-fold the level of the Opt1.5 C scenario in 2050. The scale of solar and wind power generation in the ZF scenarios differed between models due to variations in assumptions about the availability of non-fossil power sources, such as nuclear and geothermal energy, and the extent to which energy service demands could be met with electricity or hydrogen on the end-use side. In the ZF2050 scenario of MESSAGEix, solar and wind power generation in 2050 was comparable to that in the Opt1.5 C scenario of AIM. Total hydrogen generation, especially green hydrogen generation, also increased, particularly in the ZF2050 scenario of AIM, where it reached exceptionally high levels compared to typical 1.5 °C scenarios (Fig. 3d). In the Opt1.5 C scenario of MESSAGEix, fossil fuels and biomass with CCS accounted for a significant share of hydrogen generation. However, in the ZF scenarios, the phase-out of fossil fuels and the increased use of biomass in other hard-to-abate sectors resulted in hydrogen generation being dominated by green hydrogen (Supplementary Fig. 2b).

The substantial increase in power generation by 2050 was driven by the need to support the phase-out of fossil fuels on the end-use side through both direct and indirect electrification (Fig. 3a, b). According to AIM, indirect electrification via hydrogen and synthetic fuels was the primary strategy for achieving the phase-out of fossil fuels in end-use

sectors, leading to a substantial increase in green hydrogen demand, including for synthetic fuel production. As green hydrogen generation involves considerable energy losses[23], the significant increase in hydrogen generation compared to typical 1.5 °C scenarios contributed to power generation growth (Fig. 3a). In the ZF scenarios, the use of synthetic fuels increased; however, nearly all $CO_2$ sources for synthetic fuel production in 2050 were biomass-based. As a result, unlike in 2100, there was no additional power demand from DAC in 2050 (Fig. 3a and Supplementary Fig. 5a). According to MESSAGEix, both direct electrification and indirect electrification via hydrogen were key strategies for achieving the ZF goal, and increased demand for electricity and hydrogen contributed to an increase in power generation (Fig. 3b).

The rapid increase in power and hydrogen generation by the middle of the century under the ZF scenarios had a significant impact on scaling up supporting technologies such as solar and wind power, energy storage technologies, and electrolysers, particularly in the first half of the century (Fig. 3d). A comparison of annual capacity additions for these technologies between the Opt1.5 C and ZF scenarios revealed that the ZF scenarios exhibited more uneven growth rates and sharper peaks in installation in the first half of the century compared to the Opt1.5 C scenarios. The rapid upscaling observed in the ZF2050 scenarios surpassed that of the Opt1.5 C scenarios and is therefore expected to become a critical bottleneck for achieving large-scale electricity and hydrogen supply, and consequently, the phase-out of fossil fuels through direct and indirect electrification. Pushing back the target year for the full phase-out of fossil fuels to the end of the century would stabilise upscaling in the first half of the century and delay the timing of its peaks. MESSAGEix, which makes more optimistic assumptions than AIM about the availability of nuclear power and the range of energy service demands that can be met by non-hydrocarbons, showed more moderate upscaling. Energy storage upscaling may have been influenced by power system representation in the models. In AIM, green hydrogen generation absorbed surplus electricity, resulting in lower energy storage capacity compared to MESSAGEix despite the higher level of power generation.

## Emission-related outcomes

When fossil fuel phase-out is achieved, $CO_2$ emissions from the energy sector are reduced in the ZF scenarios compared to the Opt1.5 C

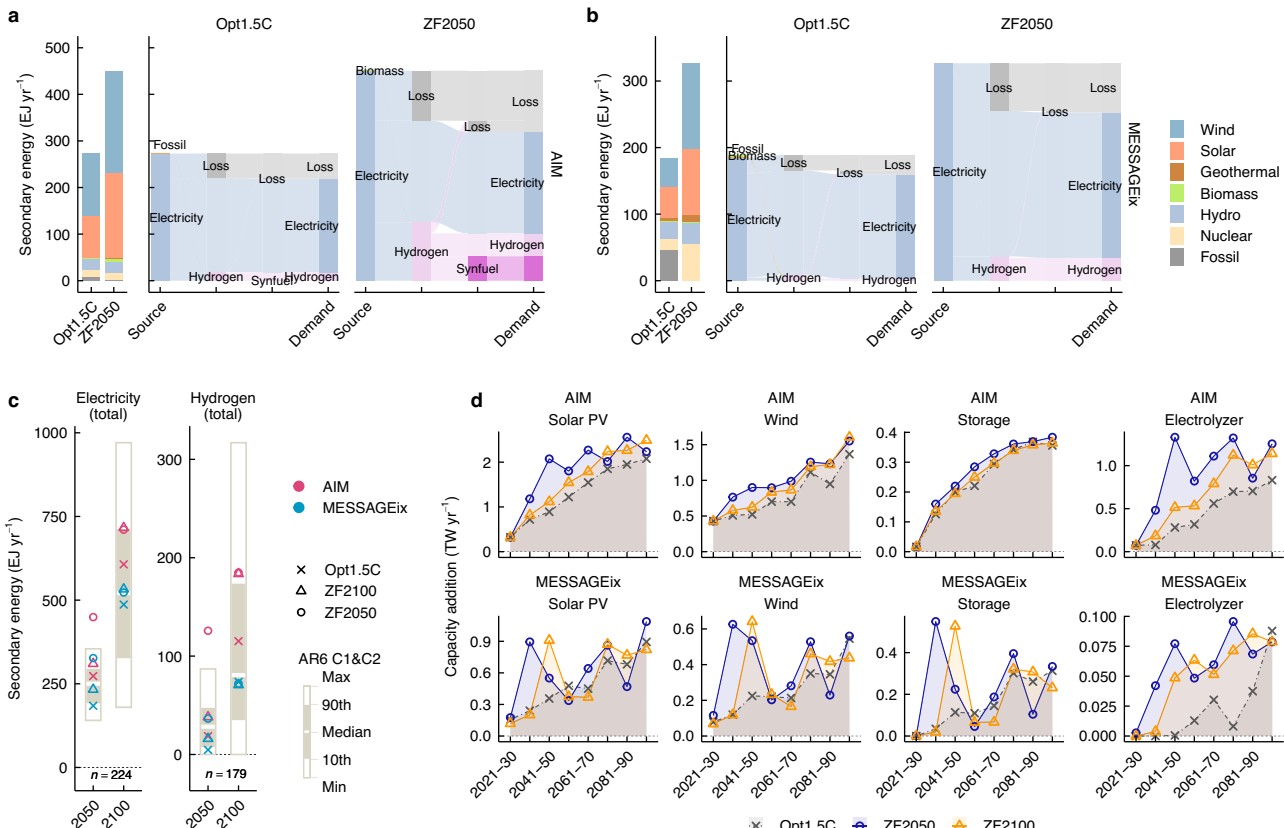

**Fig. 3 | Energy system transformation in energy supply sectors driven by a full phase-out of fossil fuels.** Power generation mixes and secondary energy flows in the Opt1.5 C and ZF2050 scenarios of AIM-Technology (**a**) and MESSAGEix-GLOBIOM (**b**) in 2050. Bar plots on the left show power generation by energy source. Diagrams on the right show secondary energy flows associated with electricity and hydrogen-based energy carrier production. Grey shading represents energy losses including conversion, storage, and distribution and trade losses, curtailment, and electricity consumption for direct air capture (DAC). Secondary energy flows in 2100 is shown in Supplementary Fig. 5. Power and hydrogen generation mixes during the period 2020–2100 is shown in Supplementary Fig. 2. **c** Total power and hydrogen generation in 2030, 2050, 2070, and 2100. Box plots illustrate power and hydrogen generation in the IPCC AR6 C1 and C2 scenarios for 2030, 2050, 2070, and 2100. **d** Annual average capacity additions for solar and wind power, energy storage technologies, and electrolyser by decade in the Opt1.5 C, ZF2100, and ZF2050 scenarios. "*n*" denotes the number of available scenarios in each category.

scenarios by 2060–2070 (Fig. 4 and Supplementary Fig. 6a). The degree of reduction from the Opt1.5 C scenario varied depending on the model and target year. $CO_2$ emissions from the energy sector in 2050 were reduced by 6–68% in AIM and by 43–89% in MESSAGEix (Fig. 4a). In 2050, residual $CO_2$ emissions from fossil fuels and industry ($CO_2$-FFI) in the Opt1.5 C scenario ranged from 12 to 16 $GtCO_2$/yr (Fig. 4b). In the ZF2050 scenario, the full phase-out of fossil fuels lowered residual $CO_2$-FFI to 2–4 $GtCO_2$/yr; residual $CO_2$ emissions from energy supply in AIM model included emissions from the utilisation of $CO_2$ captured from industrial processes. In the Opt1.5 C scenario, both models used negative emissions from bioenergy with CCS (BECCS) in the energy sector by 2050, whereas in the ZF2050 scenario, no negative emissions were deployed in the energy sector. Additionally, in MESSAGEix, which endogenously determines $CO_2$ emissions from the agriculture, forestry, and other land use (AFOLU) sector, the reduction in residual $CO_2$ emissions from the energy sector led to a decrease in negative emissions from the AFOLU sector in 2050.

In the latter half of the century, $CO_2$ emissions from the energy sector behaved differently between the AIM and MESSAGEix models, depending primarily on how emission constraints were imposed (Fig. 4a). In MESSAGEix, which applied cumulative carbon budgets for the entire century, deeper emissions reductions in the first half of the century led to higher $CO_2$ emissions from the energy sector in the ZF scenarios than in the Opt1.5 C scenario after 2070, as the saved carbon

budget was carried over. By contrast, AIM, which applied annual carbon caps, did not exhibit such a rebound, as the predefined caps in each year constrained emissions regardless of earlier reductions. In any case, cumulative $CO_2$ emissions from the energy sector between 2020 and 2100 were reduced in the ZF scenarios of both models, particularly those that achieved ZF earlier. In the ZF scenarios, cumulative $CO_2$ emissions from the energy sector between 2020 and 2100 were reduced by 2–33% compared to the Opt1.5 C scenario in AIM, and by 10–36% for that in MESSAGEix. When cumulative $CO_2$ emissions from the energy and AFOLU sectors were combined, reductions reached 2–44% in AIM and 7–32% in MESSAGEix.

The scale and approach of CCUS differed between the Opt1.5 C and ZF scenarios (Supplementary Fig. 6b). In ZF scenarios of MESSAGEix, the need for CCS decreased due to the reduction of residual emissions, resulting in smaller-scale CCS and $CO_2$ capture compared to the Opt1.5 C scenarios over the entire century. In the Opt1.5 C scenario of MESSAGEix, fossil fuel with CCS were deployed on a large scale, peaking around 2070, but its scale was significantly reduced in the ZF scenarios. By contrast, in the ZF scenarios of AIM, where CCU contributed to the phase-out of fossil fuels, CCS was decreased compared to the Opt1.5 C scenario. As a result, the overall magnitude of $CO_2$ capture remained nearly the same as in the Opt1.5 C scenario. In 2050, nearly all captured $CO_2$, including that from fossil fuels, industrial processes, and biomass, was stored underground in the Opt1.5 C

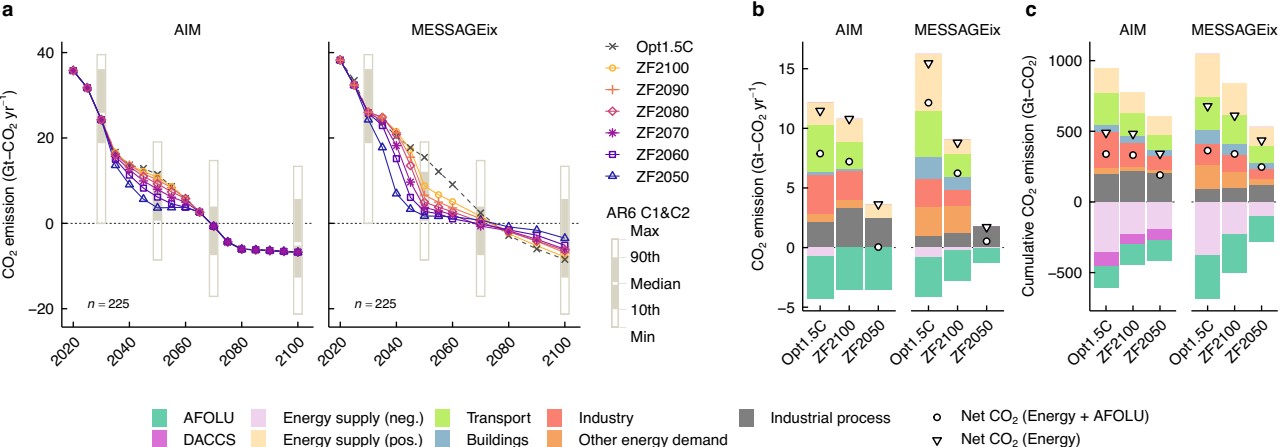

**Fig. 4 | CO$_2$ emissions in ZF energy systems. a** CO$_2$ emissions from energy and industrial processes, including negative emissions originating from direct air carbon capture and storage (DACCS). Box plots illustrate CO$_2$ emissions in the IPCC AR6 C1 and C2 scenarios for 2030, 2050, 2070, and 2100. Sectoral CO$_2$ emissions in 2050 (**b**) and cumulative sectoral CO$_2$ emissions from 2020 to 2100 (**c**) in the Opt1.5 C, ZF2100, and ZF2050 scenarios. Circles indicate net CO$_2$ emissions from the energy and agriculture, forestry, and other land use (AFOLU) sectors; triangles indicate net CO$_2$ emissions from energy sectors. "$n$" denotes the number of available scenarios in each category.

scenario, whereas in the ZF scenario, almost all of it was utilised. Even in the latter half of the century, the contribution of CCU remained limited in the Opt1.5 C scenario, whereas in the ZF scenarios, approximately 4 GtCO$_2$/yr of CO$_2$ was utilised through bioenergy with CCU (BECCU) and direct air carbon capture and utilisation (DACCU). In both the Opt1.5 C and ZF scenarios, the scale of DAC was 3–4 GtCO$_2$/yr, suggesting that, unlike power and hydrogen generation technologies, the scalability of DAC is unlikely to pose a ZF scenario-specific bottleneck.

## Challenges and opportunities

Energy transformations towards ZF energy systems highlight both challenges and opportunities from multiple perspectives (Fig. 5a–i). Phasing out even residual fossil fuel consumption in the Opt1.5 C scenarios triggered upscaling of supporting technologies, such as solar and wind power (Fig. 5a), and increased the cumulative energy investments in energy supply sectors from 2020 to 2100 by 8–31% in AIM and 8–34% in MESSAGEix (Fig. 5b). Similarly, cumulative energy investments in energy demand sectors during the same period rose by 11–27% in AIM (Fig. 5d). In AIM, an earlier target year in the ZF scenarios was associated with greater stranded capacity of coal power plants by mid-century compared to the Opt1.5 C scenario (Fig. 5c). By contrast, in MESSAGEix, which is a perfect-foresight model, the interaction between different ZF target years and the timing of investments in coal power plants was more complex. The stranded capacity showed the greatest increase in the ZF2070 scenario, and decreased in the ZF2050 scenario compared to the Opt1.5 C scenario (Fig. 5c). In the ZF scenarios, non-hydrocarbon energy penetrated more deeply on the energy demand side compared to the Opt1.5 C scenario (Fig. 5e). This result drove a shift from fossil fuel consumption technologies to electricity and hydrogen consumption technologies, which is associated with greater investments (Fig. 5d) and anticipated to result in lifestyle changes.

The phase-out of fossil fuels has a clear advantage, avoiding CCS and CDR as a consequence of deep reductions in CO$_2$-FFI compared to the Opt1.5 C scenario (Fig. 4). Cumulative geological CO$_2$ storage throughout the century was reduced by 37–46% in AIM and 52–77% in MESSAGEix compared to the Opt1.5 C scenarios (Fig. 5f). Similarly, cumulative BECCS and DACCS deployment decreased by 35–42% in AIM and 39–74% in MESSAGEix (Fig. 5g). The energy transformation required for phasing out fossil fuels may have both positive and

negative impacts on land use. In the ZF scenarios, biofuels were extensively utilised as non-fossil hydrocarbon fuels, resulting in increased primary energy supply from biomass and potentially greater pressure on the land use sector compared to the Opt1.5 C scenarios (Fig. 5h). By contrast, in MESSAGEix, which considers interactions with the AFOLU sector, stronger emission reduction efforts in the energy sector under ZF scenarios alleviated the emission reduction burden in the AFOLU sector. Consequently, the need for negative emissions through measures such as afforestation was reduced (Fig. 5i). The implications of fossil fuel phase-out for the land use sector will need to be thoroughly evaluated in future studies.

## Discussion

We conducted a model ensemble using two global energy system models to obtain robust insights into ZF energy systems and to illustrate two distinct representative pathways towards achieving such systems. Based on our results, a full phase-out of fossil fuels would require an energy system transformation that goes substantially beyond typical 1.5 °C pathways, entailing non-negligible challenges and opportunities, thereby underscoring that defossilisation should not necessarily be equated with decarbonisation. The two models employed partially different strategies for phasing out fossil fuels, most notably with respect to the degree of penetration of non-hydrocarbon energy on the energy demand side. Nonetheless, the transformation towards ZF energy systems was characterised by substantial mid-century increases in power and hydrogen generation compared to typical 1.5 °C scenarios, which could make the scalability of technologies such as solar and wind power, energy storage, and electrolysers a critical bottleneck in achieving ZF energy systems. Additionally, challenges observed in 1.5 °C scenarios, such as increased cumulative energy investments and lifestyle changes due to rapid energy system transformation[32,45], could be further amplified in ZF scenarios. The implication for stranded investments, a negative consequence of rapid energy system transformation[46,47], varied across models and scenarios, reflecting differences in investment timing and the ZF target year. By contrast, ZF scenarios showed ancillary benefits such as lower peak and end-of-century temperatures, leading to reduced climate impacts, lower reliance on CCS, and a decreased burden of emission reductions in the land use sector compared to typical 1.5 °C scenarios. Setting the target year to the end of this century, when near-ZF energy systems are achieved under typical 1.5 °C

scenarios, would reduce additional efforts but also diminish the benefits of the full phase-out of fossil fuels. Based on the fundamental premise that the full phase-out of fossil fuels is sufficient but not necessary to achieve the 1.5 °C target, it is crucial to recognise the challenges and opportunities of ZF scenarios highlighted in this study and to evaluate whether the full phase-out should be the ultimate goal of climate policy.

The increased energy investments in ZF scenarios reaffirm that if the primary objective of climate policy is to achieve the 1.5 °C goal, then typical 1.5 °C scenarios characterised by partial allowance of fossil fuels alongside abatement and removal through CCS and CDR are more cost-effective than ZF scenarios focused on the full phase-out of fossil fuels. When considering whether society should pursue the full phase-out of fossil fuels despite understanding that it is not a cost-effective 1.5 °C pathway, the quantitative and qualitative challenges and opportunities of ZF energy systems provide valuable insights for decision-making. The potential for greater increases in energy investments and lifestyle changes is likely to pose challenges to the socioeconomic viability of the full phase-out of fossil fuels. For instance, the growing reliance on electricity and hydrogen in the transport sector requires a transition from internal combustion engine vehicles to battery and fuel cell electric vehicles, which may face barriers arising from consumer preferences and behavioural habits. Alternatively, fossil-fuel phase-out pathways with limited demand-side transformations based on synthetic fuels could be pursued, but these would involve higher costs[32]. However, ZF scenarios achieve significantly greater reductions in $CO_2$-FFI compared to 1.5 °C scenarios, leading to lower peak and end-of-century temperatures, and consequently reduced climate impacts. Reduced reliance on CCS and CDR in the ZF scenarios may enhance their potential for broader societal acceptance, given the barriers to social acceptance associated with geological $CO_2$ storage[20,48,49]. Moreover, the straightforward concept of ZF scenarios, which uniformly phase out all fossil fuel consumption, may send a stronger signal to fossil fuel producers to cease investment in fossil fuel exploration, extraction, and transmission and distribution infrastructure compared to 1.5 °C scenarios, which involve a degree of ambiguity by allowing a certain amount of residual fossil fuel consumption only in hard-to-abate sectors. This simplicity may influence not only supply-side investment decisions but also facilitate understanding and implementation of ZF scenarios by the general public. The challenges and opportunities associated with the full phase-out of fossil fuels gradually diminish as the target year is extended further into the future. Some challenges and opportunities within and beyond the energy sector are complementary, where strengthening one effort can ease another. For example, if significant changes in human behaviours and lifestyles reduce energy service demand, as observed in previous studies[27,28], then the energy investments required for ZF targets could decrease. Thus, it is important to first recognise that decarbonisation and defossilisation are not necessarily equivalent, and that mitigation pathways allowing for the limited use of fossil fuels in achieving climate targets should be widely understood by the public. With this understanding, society's willingness to bear additional costs and embrace behavioural changes will determine whether a ZF energy system can become the ultimate goal, as well as when the target year for achieving it should be set, through considering the challenges and opportunities of the full phase-out of fossil fuels.

The rapid upscaling of technologies required to meet the mid-century power generation and hydrogen generation increases in ZF scenarios, compared to 1.5 °C scenarios, would likely become one of the most critical bottlenecks to achieving them. Therefore, ZF energy systems will require both policies that directly target the full phase-out of fossil fuels, such as limiting fossil fuel extraction, banning fossil fuel-consuming equipment, or phasing out fossil fuel incentives including subsidies, and complementary policies that strongly promote the market expansion and cost reduction of power and hydrogen

generation technologies. The COP28 final decision highlighted both transitioning away from fossil fuels in energy systems and tripling renewable energy capacity as global efforts for the Parties to contribute, and the results of the present study emphasise the importance of the latter approach. Such rapid upscaling of power and hydrogen generation technologies in the ZF scenarios may further exacerbate the potential mineral shortages in typical decarbonisation scenarios suggested by previous studies[50] (Supplementary Fig. 7; see also Supplementary Note 1 for details of the method). It should be noted that this assessment considers only the mineral demand associated with the energy supply sectors, following the approach used in the previous study[50]. While the main implication that the mineral shortages, particularly those related to short-term production capacity expansion, become more pronounced from the Opt1.5 C to the ZF scenarios would remain unchanged, future research should account for additional sources of mineral demand, including those from electric vehicles. CCUS and CDR are often debated with respect to their potential to delay the phase-out of fossil fuels. The COP28 final decision acknowledged the contributions of abatement technologies and transitional fuels, such as fossil power plants with CCS, blue hydrogen, and ammonia. While some 1.5 °C scenarios, including the Opt1.5 C scenario in MESSAGEix, suggested potential contributions from these bridging technologies, their role was limited in the ZF scenarios, including the ZF2100 scenario. Therefore, allowing fossil CCUS technologies could be regarded as a loophole, and may not align with the full phase-out of fossil fuels as the ultimate goal of global climate change mitigation policy. However, unlike fossil CCUS technologies, the contribution of non-fossil CCUS in the ZF scenarios indicates they are no longer installed as an excuse to retain fossil fuels, but rather as a serious contribution to climate change mitigation. As highlighted by the COP28 final decision, the phrase "transitioning away from fossil fuels in energy systems, in a just, orderly and equitable manner" underscores the importance of addressing the heterogeneous regional implications of a full phase-out of fossil fuels. In line with recent discussions on a just transition, which includes, for example, fairness in energy and climate decision-making processes and fostering of both international cooperation and coordinated multilateral actions[51], the ZF scenarios should also emphasise mitigating the effects of energy transformations and envisioning an equitable decarbonised world[52]. Notably, as the ZF scenarios require a more rapid and deeper phase-out of fossil fuels than typical 1.5 °C scenarios, greater attention will need to be given to countries that are highly dependent on fossil fuel revenues and have limited capacity for transition[53]. While detailed national- and regional-scale analyses are beyond the scope of this study, the ZF scenarios showed that international fossil fuel trade, which persisted to some extent even by the end of the century in the Opt1.5 C scenarios, was phased out and replaced by expanded international trade in biomass and hydrogen-based energy carriers (Supplementary Fig. 8). Although these findings are incomplete, they suggest significant impacts on current fossil fuel-exporting countries, highlighting the importance of carefully determining when and where to phase out fossil fuels beyond the uniform assumptions employed in this study. In addition, complementary policies supporting a just and equitable phase-out of fossil fuels will be essential to secure their cooperation[53].

This study had several limitations. First, the representation of energy demand sectors by the models may have overlooked some bottlenecks associated with the full phase-out of fossil fuels from the energy demand side. These include sectors such as steel[19], chemicals[54], and aviation[55], whose representation was simplified in one or both of the models used in this study. Nonetheless, our scenario analysis likely provides sufficient qualitative insights into potential energy system transformations, along with associated challenges and opportunities, for reducing fossil fuel consumption to nearly zero. Second, while the models used in this study provided a broad assessment of the

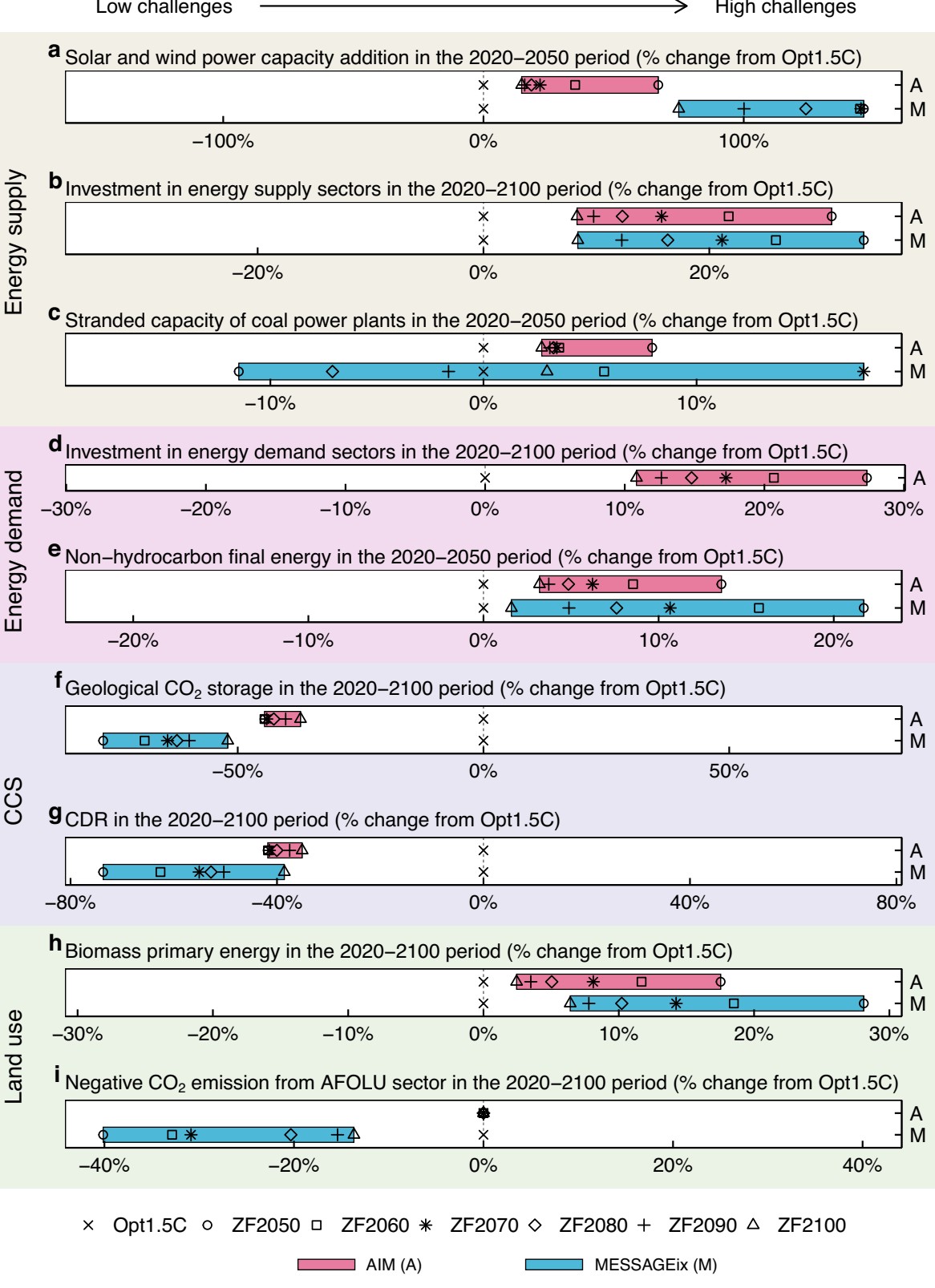

**Low challenges** ⟶ **High challenges**

**a** Solar and wind power capacity addition in the 2020–2050 period (% change from Opt1.5C)

**b** Investment in energy supply sectors in the 2020–2100 period (% change from Opt1.5C)

**c** Stranded capacity of coal power plants in the 2020–2050 period (% change from Opt1.5C)

**d** Investment in energy demand sectors in the 2020–2100 period (% change from Opt1.5C)

**e** Non–hydrocarbon final energy in the 2020–2050 period (% change from Opt1.5C)

**f** Geological $CO_2$ storage in the 2020–2100 period (% change from Opt1.5C)

**g** CDR in the 2020–2100 period (% change from Opt1.5C)

**h** Biomass primary energy in the 2020–2100 period (% change from Opt1.5C)

**i** Negative $CO_2$ emission from AFOLU sector in the 2020–2100 period (% change from Opt1.5C)

Energy supply · Energy demand · CCS · Land use

× Opt1.5C   ○ ZF2050   □ ZF2060   ✳ ZF2070   ◇ ZF2080   + ZF2090   △ ZF2100

AIM (A)     MESSAGEix (M)

interrelationships within the energy system under a full phase-out of fossil fuels, they also exhibited limitations in their representation of power systems. Both models have introduced innovative approaches to capture challenges related to integrating variable renewable energy (VRE), such as considering hourly dispatch for selected representative days during technology selection and incorporating constraints on system flexibility and capacity reserves[12,56]. However, other models are specifically designed for the detailed analysis of power systems by narrowing their focus to specific regions and sectors, and the models used in the present study fall short of such specialised models in terms of spatiotemporal resolution[57,58]. While the qualitative findings of our study are unlikely to be affected, a deeper understanding of the power system transformation required to support the increased power generation in the ZF scenarios will be necessary in future analyses. Third,

**Fig. 5 | Challenges and opportunities for achieving the full phase-out of fossil fuels from multiple perspectives. a** Percent changes in cumulative capacity additions of solar and wind power from 2020 to 2050 relative to the Opt1.5 C scenario. **b** Percent changes in cumulative energy investment in energy supply sectors from 2020 to 2100, discounted by 5% per year relative to the Opt1.5 C scenario. **c** Percent changes in the annual average stranded capacity of coal power plants from 2020 to 2050 relative to the Opt1.5 C scenario, where stranded capacity is defined as the unused capacity of power plants in each period. **d** Percent changes in cumulative energy investments in energy demand sectors from 2020 to 2100,

discounted by 5% per year relative to the Opt1.5 C scenario. Only AIM provided this indicator. **e** Percent changes in cumulative non-hydrocarbon energy final consumption from 2020 to 2050 relative to the Opt1.5 C scenario. **f** Percent changes in cumulative geological $CO_2$ storage from 2020 to 2100 relative to the Opt1.5 C scenario. **g** Percent changes in cumulative CDR via BECCS and DACCS from 2020 to 2100 relative to the Opt1.5 C scenario. **h** Percent changes in cumulative primary energy supply from biomass 2020 to 2100 relative to the Opt1.5 C scenario. **i** Percent changes in cumulative negative $CO_2$ emissions from the AFOLU sector from 2020 to 2100 relative to the Opt1.5 C scenario.

we acknowledge that the two models used to explore ZF energy systems in this study may not have fully captured the entire solution space. Intercomparison of 1.5 °C scenarios using two models, MESSAGEix and REMIND (REgional Model of Investment and Development) has been conducted previously[8]; however, a broader model ensemble involving more models would be desirable in the future. Future research could further analyse the sustainability implications of ZF energy systems, including the land use impacts and their potential heterogeneous regional effects.

## Methods

### Energy system models

We employed two global energy system models in this study, which are briefly described. AIM-Technology is a recursive dynamic energy system model covering all regions of the world. AIM-Technology includes 31 regions and various energy sectors, including industry (steel, cement, and other industries), buildings (residential and commercial), transportation (passenger and freight), and energy supply (power generation, hydrogen generation, fossil extraction, and others). The model solves a linear programming problem at each time step to estimate the status of deployment, operation of energy technologies, and associated emissions by minimising the total energy system cost, which is represented as the sum of capital costs, operation and maintenance costs, energy costs, and emission costs for each technology. Constraints ensure that system requirements such as exogeneous energy service demands and emission caps are met. AIM-Technology operates with one-year time steps from 2005 to 2050 and five-year time steps from 2055 to 2100. Details of the model structure and mathematical formulation are provided in refs. 26,[12].

AIM-Technology is a technology-rich model that features detailed representations of energy supply and demand technologies across various energy sectors and sub-sectors. A comprehensive list of its technology options is available at the AIM-Technology documentation page (https://kenoshiro.github.io/AIM-Technology-doc/). AIM-Technology accounts for the extraction of fossil fuels such as coal (hard coal and lignite), crude oil, and natural gas. Resource potential is categorised into 12 different grades based on resource type and extraction cost. The supply potential of dedicated energy crops is classified into eight grades according to production costs. These costs are exogenously determined based on ref. 59. Other bioenergy supply potential including agriculture residue, forest residue, municipal solid waste and manure is obtained from GEA[60]. AIM-Technology includes an hourly resolution dispatch module, enabling detailed consideration of the variability in power supply from VREs such as solar and wind power, as well as variability in power demand. In this study, to account for seasonal variation in power supply from VREs and power demand, the model analysed the power supply–demand balance on an hourly basis for 12 representative days, one from each month. AIM-Technology assumes inelastic energy service demands. Certain energy services are modelled with technologies that have different levels of efficiency, and improvements in energy efficiency through the substitution by higher-efficiency technologies are determined endogenously. Fuel switching is represented by assuming different sets of available end-use technologies for each energy service at the subsector

level. AIM-Technology models the production of liquid and gaseous synthetic hydrocarbon fuels from hydrogen and captured $CO_2$. It includes hydrogen generation through electrolysis, biomass and coal gasification, and natural gas steam reforming. Furthermore, it enables $CO_2$ capture from large emission sources, such as power and hydrogen generation, oil refining, biomass liquefaction, steel and cement production, and furnaces, as well as from DAC. AIM-Technology accounts for trade in coal, crude oil, natural gas, oil products, biomass (solid and liquid), and hydrogen-based energy carriers (ammonia, synthetic fuels, and methylcyclohexane). AIM-Technology does not consider electricity trade.

MESSAGEix–GLOBIOM[42,43] soft-links the energy system model MESSAGEix[41,61] with the land use model GLOBIOM[62,63]. MESSAGEix is a perfect foresight energy system model covering all regions of the world. It includes 11 regions and various energy activities, such as energy extraction, energy conversion (electricity, heat, liquid fuels, gaseous fuels, and hydrogen), and final energy consumption (industry, buildings, and transportation). MESSAGEix solves a linear programming problem to estimate the least-cost portfolio, minimising total system costs expressed as the sum of capital costs, operation and maintenance costs, and costs for emissions and land use, while considering given service demands and emission constraints. Detailed formulations are available at the MESSAGEix documentation page (https://docs.messageix.org/). For time slices set at 5-year intervals from 2025 to 2050 and ten-year intervals from 2055 to 2110, MESSAGEix optimises the total discounted system costs as the sum across these time slices. MESSAGEix estimates the macro-economic demand response based on energy system and service costs through iterative calculations with the single-sector macroeconomic module MACRO[62]. GLOBIOM, which is a partial-equilibrium land use model, provides MESSAGEix with information on land-use dynamics, such as the potential and costs of bioenergy and the opportunities and expenses for emission reductions in the AFOLU sector. To reduce computational costs, rather than running the full GLOBIOM model iteratively, the MESSAGEix model adopts a GLOBIOM emulator, built on an ensemble of land-use pathways provided by GLOBIOM. Each pathway reflects population-driven developments from both land-use and nutritional perspectives, resulting in different levels of bioenergy supply across a range of carbon prices.

A detailed list of technology options available in MESSAGEix is available at the MESSAGEix documentation page (https://docs.messageix.org/projects/models/en/latest/). MESSAGEix covers extractions of coal, lignite, crude oil, and natural gas, with the potential of each resource graded according to varying extraction costs. MESSAGEix considers the reliability and flexibility requirements of the power system not by explicitly accounting for hourly power supply and demand, but by imposing a constraint that ensures sufficient dispatchable generator capacity in each time slice. In this study, no iteration between MESSAGEix and MACRO was carried out, and consequently, inelastic useful energy demand was assumed. The magnitude of the macroeconomic response of final energy demand in MESSAGEix-MACRO is shown in Supplementary Fig. 9. Energy efficiency improvements are modelled statically as exogenous trends over time and are implicitly embedded within the demand projections. The

extent of fuel switching is represented by imposing share constraints. The version of MESSAGEix used in this study includes hydrogen generation through coal and biomass gasification, natural gas steam reforming, and electrolysis, but does not include synthetic hydrocarbon production from captured $CO_2$ and hydrogen. Thus, all captured $CO_2$ is assumed to be stored underground. While $CO_2$ capture from large point sources is considered, DAC is not included. MESSAGEix assumes trade in solid, liquid and gaseous fuels, electricity, and liquid hydrogen.

## Scenarios

In this study, multiple scenarios were modelled to understand diverse transition pathways for ZF energy systems. The ZF scenarios are labelled based on the target year for achieving full phase-out of fossil fuels (for example, ZF2050 indicates the complete elimination of fossil fuels by 2050), with constraints imposed on the upper limit of the fossil fuel primary energy supply, along with $CO_2$ emission constraints consistent with the 1.5 °C target. The target years for ZF were set at 10-year intervals from 2050 (ZF2050) to 2100 (ZF2100). We employed a typical 1.5 °C scenario (Opt1.5 C) for comparison, without imposing the upper limits of primary supply of fossil fuels. The socioeconomic conditions are based on the middle-of-the-road Shared Socioeconomic Pathway (SSP2)[64].

In AIM-Technology, the upper limits on the primary supply of fossil fuels each year were determined by the primary supply in the same year in the Opt1.5 C scenario and the reduction rate from the Opt1.5 C scenario. First, the primary supply of fossil fuels in the Opt1.5 C scenario without fossil phase-out constraints was obtained. The upper bounds of the primary supply of fossil fuels for ZF scenarios were obtained by multiplying this supply with the scenario-specific reduction pathways, expressed as values relative to the Opt1.5 C scenario. The primary supply of fossil fuels was reduced linearly, starting at 100% of the Opt1.5 C scenario level in 2030 and declining to 0% of the Opt1.5 C scenario level by the target year. In MESSAGEix–GLOBIOM, which uses intertemporal optimisation, the pathways for phasing out fossil fuels are determined endogenously with greater flexibility compared to AIM-Technology, which is more myopic. Specifically, the upper limits on primary supply of fossil fuels each year are set to the levels for the same year in the Opt1.5 C scenario before the target year, and to zero thereafter. In both models, the upper limits of fossil fuels were imposed by each region and fuel type (coal, crude oil, and natural gas).

In the mitigation scenarios, we imposed $CO_2$ emission constraints consistent with 1.5 °C temperature stabilisation, specifically limiting cumulative $CO_2$ emissions across all sectors from 2018 to 2100 to 500 $GtCO_2$[10]. First, in AIM-Technology, which is a recursive dynamic model, upper bounds on annual $CO_2$ emissions were applied. The target emissions in AIM-Technology were $CO_2$ emissions from energy and industrial processes. Since AIM-Technology focuses only on the energy sector, the annual $CO_2$ emissions from energy and industrial processes, calculated under the condition of limiting cumulative $CO_2$ emissions across all sectors to 500 $GtCO_2$, were obtained from the integrated assessment model AIM-Hub and used as emission constraints. Additionally, $CO_2$ emissions from the AFOLU sector were fixed based on the outputs of AIM-Hub. In MESSAGEix–GLOBIOM, which is an intertemporal optimisation model, an upper limit was imposed on cumulative $CO_2$ emissions from all sectors over the period 2018–2100. The target emissions in MESSAGEix–GLOBIOM include $CO_2$ emissions from all sectors. Unlike AIM-Technology, MESSAGEix–GLOBIOM endogenously determines $CO_2$ emissions from the AFOLU sector.

We used the C1 and C2 scenarios obtained from the AR6 scenario database[14] as examples of typical 1.5 °C energy systems for comparison with the ZF scenarios. We used the World v1.1 dataset (AR6_Scenarios_Database_World_v1.1), which was downloaded from the AR6 Scenario Explorer (https://data.ece.iiasa.ac.at/ar6/) on 28 October 2025.

The C1 scenarios limit the temperature increase to 1.5 °C, with no or limited overshoot, with a 50% likelihood. The C2 scenarios return warming to 1.5 °C with a 50% likelihood after a high overshoot. The AR6 scenario database is described in IPCC[65]. Some indicators analysed in this study could not be directly obtained from the scenario database and were therefore calculated based on reported indicators. Regarding the shares in total final energy consumption shown in Fig. 2b, the final energy consumption of hydrocarbon fuels such as fossil fuels and biofuel was not disaggregated in detail for many scenarios. Therefore, it was calculated by fuel type (solid, liquid, and gaseous) as follows. Solid fuels were calculated directly using the final energy consumption of solid coal ("Final Energy|Solids|Coal") and solid biomass ("Final Energy|Solids|Biomass"). Liquid fuels were disaggregated based on the secondary energy mix of liquid fuels in each scenario to approximate the final energy consumption of liquid fossil fuels and biofuels. Specifically, the share of liquid biomass ("Secondary Energy|Liquids|Biomass") in the total secondary liquid energy ("Secondary Energy|Liquids") was multiplied by the total final energy consumption of liquid fuels ("Final Energy|Liquids") to estimate the final energy consumption of liquid biofuels. The remaining portion of the final energy consumption of liquid fuels was attributed to liquid fossil fuels. Scenarios that did not report secondary energy for liquid biomass ("Secondary Energy|Liquids|Biomass") were excluded from this calculation. Gaseous fuels were calculated in the same manner as liquid fuels. The share of gaseous biomass ("Secondary Energy|Gases|Biomass") in total secondary gaseous energy ("Secondary Energy|Gases") was multiplied by the total final energy consumption of gaseous fuels ("Final Energy|Gases") to estimate the final energy consumption of gaseous biofuels. The remaining portion of the final energy consumption of gaseous fuels was attributed to gaseous fossil fuels. In cases where secondary energy for gaseous biomass ("Secondary Energy|Gases|Biomass") was not reported, it was replaced with zero during the calculation.

## Data availability

The scenario data generated in this study have been deposited in the Zenodo repository (https://doi.org/10.5281/zenodo.19364675). The data of the MESSAGEix-GLOBIOM baseline scenario used in this study have been deposited in the Zenodo repository (https://doi.org/10.5281/zenodo.10514052).

## Code availability

The source code used for figure production from the scenario data is provided in the GitHub repository (https://github.com/shotaromo/YSSP2024-ZF; https://doi.org/10.5281/zenodo.19364675). The source code of the AIM-Techology model is available at the GitHub repository (https://github.com/KUAtmos/AIMTechnology_core; https://doi.org/10.5281/zenodo.8401421)[66]. Some input data used in the AIM-Technology model are derived from proprietary datasets and are not publicly available due to licensing restrictions. They are available from the corresponding author, subject to the terms and conditions of the data providers. The source code of the MESSAGEix-GLOBIOM model is available at the GitHub repository (https://github.com/iiasa/message_ix; https://doi.org/10.5281/zenodo.14892418)[67]. The MESSAGEix scenario and model data used in this study, including all sets, parameters, variables, and equations, are derived from MESSAGEix-GLOBIOM 1.1 R11 (https://doi.org/10.5281/zenodo.10514052) and have been deposited in the Zenodo repository (https://doi.org/10.5281/zenodo.17472924).

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

## Acknowledgements

Part of this research was developed in the Young Scientists Summer Program at the International Institute for Applied Systems Analysis, Laxenburg (Austria), with support from the National Member Organization. S.M. acknowledges the Madume Research Encouragement Prize Award. Funding. S.F. discloses support for the research of this work from the Japan Science and Technology Agency (JST) as part of the Adopting Sustainable Partnerships for Innovative Research Ecosystem (ASPIRE, grant number JPMJAP2331) and the Sumitomo Electric Industries Group CSR Foundation. S.M. discloses support for the research of this work from the Support for Pioneering Research Initiated by the Next Generation, presented by the Division of Graduate Studies, Kyoto University (JST SPRING, grant number JPMJSP2110). S.J., O.F., and V.K. disclose support for the research of this work from the European Union's Horizon Europe Research and Innovative Action Programme under Grant Agreement No. 101137582 (HYway) and Grant Agreement No. 101183367 (NEWPATHWAYS).

## Author contributions

S.M., S.J., V.K. and S.F. conceptualized the research. S.M. contributed to scenario design. All authors participated in the interpretation of the results. K.O. and O.F. developed the model. S.M. conducted the analysis, created the figures, and wrote the first draft of the paper. S.M., S.J., V.K., T.H. and S.F. contributed to the final manuscript.

## Competing interests

T.H. is affiliated with Toyota Motor Corporation, which did not provide specific support for this paper. The other authors declare no competing interests.
