## [Transparent Peer Review file · Nature Communications]

Challenges and opportunities of the full phase-out of fossil fuels under the 1.5°C goal

Corresponding Author: Mr Shotaro Mori

Version 0:

Reviewer comments:

Reviewer #1

(Remarks to the Author)

1. Key mineral resources should be considered

According to the current scenario, the demand for electrolytic cells and solar photovoltaics will shift towards the demand for upstream key mineral resources, such as lithium, indium, etc. The current global reserves and production capacity may not be able to meet the demands. Suggest the author to supplement the quantitative consideration of key mineral resources upstream

2. The figures can be considered for optimization

For example, Fig.1a displays AIM, MESSAGEix, and AR6 data simultaneously, with insufficient color differentiation and a lack of prominent comparison in the proportion of fossil fuels. Suggest adjusting the color to make the grouping clearer and highlighting the key points through additional annotations. Additionally, Fig.1, Fig.3 contain too much information, it is recommended to structure them appropriately

3. Suggest adding the latest literature

Key achievements after 2024 have not been cited, such as The IEA's "clean energy transitions programme 2024" proposes the goal of "zero growth of fossil fuels by 2035", which is in line with the ZF2050 path and needs to be cited to enhance policy timeliness; Besides, most of the references for IAM's scenario research are from 10 years ago.

4. There is room for further improvement in the practical value of the article

Firstly, if interdisciplinary perspectives are taken into account, the practical value of this article will be enhanced. At present, there is no cited research on "energy justice", which makes it difficult to provide theoretical support for "just transition"; In addition, ignoring behavioral economics literature, consumer behavior modeling was not included in the discussion. Besides, at present, the article does not propose an implementation roadmap and does not consider the dynamic possibilities of industrial economy and technological development.

5. Lack of regional adaptability plan

This study focuses on global models without considering regional differences, and does not cross validate cross regional models with current models such as the Chinese MARKAL model and the European TIMES model. Suggest giving appropriate consideration to regional differentiation and proposing customized solutions for representative regions

6. Some statements can be considered more rigorous and complete.

P2, L38-41. It is difficult to say the final decision language is "weaker" or not, because there must be not only scientific consideration but also geopolitical ones.

P2, L42-43. It could be better to mention in which aspects or in which ways, the energy transformation pathways could be different.

(Remarks on code availability)

Reviewer #2

(Remarks to the Author)

The manuscript is innovative and provides novel insights as to the role of COP28 outcomes as to the role of zero fossil [fuel] future energy systems and their differences as to more conventional net-zero carbon scenarios within the current body of literature.

This paper has the potential to become a seminal paper. It is incremental in methodological nature, but insightful and novel in the scenarios analyzed and the tone they potentially might set within the broader energy systems modelling and integrated assessment modelling community and more significantly, the policy bodies that consume these insights in their policy decision making process.

I did not find any Major flaws. Most of my comments are minor and suggestions in nature.

The paper is worthy of publication in my view. It could be published as is, but I expect my comments will aid in some clarity for the reader as minor revisions.

As always, some of my questions are answered by details, text, data and charts within the Extended Information section. However, one or two sentences should migrate forward into the main body. {GAS w/CCS and DACCS}

A point of note to the lead author. I believe you are an early stage career researcher, a PhD candidate that attended the IIASA YSSP - If this is the case, Excellent work - well done - keep it up.

The following comments are my main points for your consideration and minor revisions and are the same as per the attached excel file:

ID Line Number Reviewer Comment

1 94 "Robust" - consider whether or not it is appropriate to use the term "Robust" to avoid confusion with "Robust Optimisation"

2 139-146 I'm wondering about the demand response options at this point, either through demand elasticities, energy service demand efficiencies or fuel switching. It seems a gap at this point

3 154 In figure 1 - it's a great chart, information dense and spatially efficient - in panel f, I'm left wondering what GAS w/CCS is being used for in the Opt1.5C scenario. Perhaps add a description to the caption

4 209 Interesting finding with regard to Biomass mid century. Is there a trade off between land use availability and population dynamics in MESSAGE and AIM? Does that impact on the supply curve later in the model horizon?

5 214 What are the end of horizon effects within the models of having this policy in the final period. Would it impact on the robustness of signals to other sectors within the models (land use and climate?)

6 219 It may be addressed later - I'm wondering what are the costs and benefits of rapid scale up in energy crop production. Land use impacts? Biodiversity losses?

7 290 Figure 3 panel d - Very Interesting - the volatility in net-capacity increase raises obvious questions about the cashflow volatility within the technology supply chain, volatile labour markets and resulting price volatility for technologies - I wonder would showing net new capacity additions be more instructive and remove the retiring of end of life and fully depreciated technologies from the volatility. It might make it easier to understand this volatility

8 290 Following comment 7 - is there more or less volatility at the regional levels?

9 305 Seems strange to talk about future scenarios in the past tense - I'm unsure of Journal guidelines but perhaps consider discussing scenarios in the continual present tense "when fossil fuel phase-out is achieved,..."

10 327 An annual carbon constraint is not a "carbon budget", it's a "carbon cap" and should be called such to avoid confusion

11 321 The interaction between the two constraints, zero fossil and carbon budgets is interesting. In MESSAGE the ZF constraints binds before 2070 I expect and then the solution relaxes under a relatively larger carbon budget than a linear trajectory. I expect the marginal cost of carbon of those constraints are also displaying volatile non-linear behaviour. Can you confirm that the discount rates and hurdle rates are static with time and do not change around 2070?

12 321 Am I correct in remembering that AIM has dynamic recursive foresight and MESSAGE has perfect foresight? How does this and the annualised Carbon Cap (AIM) vs the Carbon Budget (MESSAGE) explain the post 2070 dynamics divergence in figure 4.a?

13 377 I see Energy Demand (5.d) is skipped in this paragraph, but I'd welcome a sentence describing the demand sector investments within AIM and what they mean. A 10-20% increase in demand investments seems considerable and worthy of description.

14 451 Is it worth mentioning the simplicity of the policy messaging around zero fossil fuel and how it might be easier to understand and implement for the general public?

15 733 Within the document I believed that DAC is included as a technology in both models, but then in the EI it is stated that DAC is not in MESSAGE-IX. This should be clarified in the main body text.

16 779 "We used the world V1.1 dataset" - for what? I don't understand this sentence

17 868 ED Figure 1-Panelf - shows a stark result in the differences for Natural Gas medium term outlook to 2060 in the high Opt1.5C case to considerably low Methane Gas in the ZF scenarios. Interesting - no action required. However, this does link back to my previous question about what is GAS w/CCS being used for in Opt1.5C? Industry?

(Remarks on code availability)

I briefly reviewed the multiple Zenodo repositories and github repositories for the core MESSAGE-IX and AIM main branches.

It would take me some days/weeks to figure out if the code paper results are reproducible with the files provided.

The models are both open source and on github - this is excellent and following best practice in my view.

However, it is unclear to me if the files provided in the zenodo archives are simply model results files and post-processing scripts and not input parameters, or model constraints specific to this paper. (Carbon budgets, Carbon Caps and ZeroFossil constraint formulations)

I think it would be useful to archive the branch of the model version on github [& zenodo if desired] to make it easier to assess the replicability - i.e. keep all the model files needed to reproduce this paper in one location on github. This is common practice in some earth systems and data science communities, but not yet in IAM/ESOM communities.

Version 1:

Reviewer comments:

Reviewer #1

(Remarks to the Author)

Thanks for the revision of the manuscript and responses to previous comments. I still have some minor questions and comments:

General:

(1) The quantitative analysis of model results is quite detailed. However, more important conclusions need to be extracted, to further improve the practical value of the article. For example, with the solid quantitative analysis result, and all those recognized opportunities and challenges, is it possible to answer the question whether on earth the full phase-out of fossil fuels scenario is feasible or not?

(2) Please explain about the reason/meaning of employing two IAM models in this study.

Details:

(3) Line 40-43, please add the reference(s).

(4) Line 108-114, what is the difference between the Opt1.5C scenario developed in this study and the C1 and C2 scenarios included in AR6? The answer may solve the concern about the necessity of the Opt1.5C scenario.

(5) Line 119-120, please check the data of text and figures. I read from the Fig. 1a that the proportion of fossil fuels are more than 40% and 60% in 2050 for the Opt1.5C scenario in AIM and MESSAGEix, respectively, which is different from 35% and 54% that mentioned in the text.

(6) Fig 3d. under ZF scenarios, especially in the ZF2050 scenarios, what causes the fluctuations of these technologies?

(Remarks on code availability)

Version 2:

Reviewer comments:

Reviewer #1

(Remarks to the Author)

Thanks very much for the revisions and responses. I think the manuscript is now can be accepted.

(Remarks on code availability)

We appreciate the reviewers' comments and for the opportunity to submit a revised manuscript. The manuscript has been revised thoroughly according to both editorial and reviewer comments. This document contains point-by-point responses to each comment in blue. All changes in the revised manuscript are highlighted in yellow.

Reviewer #1 (Remarks to the Author):

1. Key mineral resources should be considered

According to the current scenario, the demand for electrolytic cells and solar photovoltaics will shift towards the demand for upstream key mineral resources, such as lithium, indium, etc. The current global reserves and production capacity may not be able to meet the demands. Suggest the author to supplement the quantitative consideration of key mineral resources upstream.

Response:

Thank you for your valuable comment, with which we are in complete agreement. A potential shortage of key minerals could become a bottleneck in our ZF scenarios, as highlighted in recent studies¹.

Accordingly, we have conducted a supplementary analysis focusing on copper, lithium, nickel, cobalt, graphite, and rare earth elements (praseodymium, neodymium, terbium and dysprosium), which were identified as key energy transition minerals in the IEA's "Global Critical Minerals Outlook 2025"². Following the approach of Wei et al. (2025)¹, we estimated the cumulative demand for key minerals based on technology capacity additions and corresponding mineral intensities, as described in the caption of Supplementary Note 1.

As a result, the cumulative demand for all key minerals increases in the ZF scenarios compared with the Opt1.5C scenarios, and this increase is particularly pronounced in the ZF scenarios of the MESSAGEix model, where substantial near-term capacity additions of solar and wind are required relative to the Opt1.5C scenario. The main implication of this study, which aims to explore the challenges and opportunities associated with achieving a full phase-out of fossil fuels, is that the potential barriers related to mineral shortages, particularly those associated with short-term production capacity expansion rather than reserves, become more significant in the ZF scenarios than in the Opt1.5C scenario. Note that mineral demand from electric vehicles and grid reinforcement was not considered in Wei et al. (2025)¹, and in our estimation, we did not perform a detailed analysis of secondary supply and reuse. Therefore, these values alone cannot be used to draw a conclusive judgment regarding potential mineral shortages, highlighting the need for further research that takes these factors into account. However, they are sufficient to offer quantitative insights into the additional upstream demand for key minerals and the associated supply constraints implied by the capacity additions from the Opt1.5C to the ZF scenarios, as shown in Fig. 3.

Supplementary Fig. 7 | Key mineral demand from energy supply technologies. Cumulative demand for key minerals, including copper, lithium, nickel, cobalt, graphite and rare earth elements (praseodymium, neodymium, terbium, and dysprosium), in the ZF2050 and ZF2100 scenarios relative to the Opt1.5C scenario. The estimates cover mineral demands from 13 power generation technologies (coal with/without CCS, oil with/without CCS, gas with/without CCS, nuclear, hydro, geothermal, biomass with/without CCS, solar, and wind), as well as electricity storage and electrolysers. Mineral demand was estimated using the technology-specific mineral intensities (e.g., $t\ GW^{-1}$) reported in Wei et al. (2025)³ together with the capacity additions (e.g., $GW\ yr^{-1}$) in each scenario. Further details of the methodology are provided in Supplementary Note 1.

We have added the following discussion and included Supplementary Fig. 7 in the Supplementary Information.

Page 17, line 498-507

Such rapid upscaling of power and hydrogen generation technologies in the ZF scenarios may further exacerbate the potential mineral shortages in typical decarbonization scenarios suggested by previous studies⁵⁰ (Supplementary Fig. 7; see also Supplementary Note 1 for details of the method). It should be noted that this assessment considers only the mineral demand associated with the energy supply sectors, following the approach used in the previous study. While the main implication that the mineral shortages, particularly those related to short-term production capacity expansion, become more pronounced from the Opt1.5C to the ZF scenarios would remain unchanged, future research should account for additional sources of mineral demand, including those from electric vehicles.

Furthermore, the detailed method is provided in Supplementary Note 1, as described below.

Supplementary Information, Page 10

We estimated the demand for key minerals, including copper, lithium, nickel, cobalt, graphite, and

rare earth elements (praseodymium, neodymium, terbium and dysprosium), following the data and approach of Wei *et al.* (2025)³. The coverage of these key minerals was defined in accordance with the key energy transition minerals specified in the IEA's *Global Critical Mineral Outlook 2025*⁶. Wei *et al.* (2025)³ provide technology-specific mineral intensity data (e.g., $t\ GW^{-1}$) for 13 power generation technologies (coal with/without CCS, oil with/without CCS, gas with/without CCS, nuclear, hydro, geothermal, biomass with/without CCS, solar, and wind) as well as for electricity storage and electrolysers. For solar power, electricity storage, and electrolysers, the data are further disaggregated into multiple technology types, and individual mineral intensities and future market shares are provided for each. By combining these mineral intensities with annual capacity additions (e.g., $\text{GW}\ \text{yr}^{-1}$), we estimated the corresponding mineral demand (e.g., $t\ \text{yr}^{-1}$) for each scenario. It should be noted that, as discussed as one of the limitations in Wei *et al.* (2025)³, their technology-specific mineral intensity data do not cover several technologies such as electric vehicles and transmission grids. In addition, this study does not account for mineral recycling, and therefore the estimated mineral demand represents the total amount, including the portion that could be supplied by recycled minerals.

References:

1. Wei, Y.-M. *et al.* Navigating energy transition solutions for climate targets with minerals constraint. *Nat. Clim. Change* **15**, 833–841 (2025).
2. IEA. *Global Critical Minerals Outlook 2025*. (2025).

2. The figures can be considered for optimization

For example, Fig.1a displays AIM, MESSAGEix, and AR6 data simultaneously, with insufficient color differentiation and a lack of prominent comparison in the proportion of fossil fuels. Suggest adjusting the color to make the grouping clearer and highlighting the key points through additional annotations. Additionally, Fig.1, Fig.3 contain too much information, it is recommended to structure them appropriately

Response:

Thank you very much for your suggestion.

First, to improve the readability of Fig. 1a, we made the following adjustments:

- We changed the colours for AIM, MESSAGEix, and AR6 to enhance clarity.
- We added annotations to indicate the direction of the energy mix transition associated with the fossil fuel phase-out.
- We differentiated the point sizes between ZF2050/Opt1.5C and the other scenarios to highlight the comparison between these two scenarios.
- We unified the shapes of AR6 C1 and C2 points, since distinguishing between them is not essential in this figure, thereby reducing unnecessary information.

Fig. 1 | Fossil fuel phase-out and energy system transformation.

Second, to reduce the amount of information in Fig. 3 without affecting the discussion in the main text, we made the following adjustments:

- For Fig. 3c, we focused on the key scenarios (Opt1.5C, ZF2100, and ZF2050) and the years 2050 and 2100.
- Similar to Fig. 1a, we revised the colour coding in Fig. 3c to improve clarity.
- Following the changes to the model-based colours in Fig. 3c, we modified the scenario-based colours in Fig. 3d so that they do not overlap.

Fig. 3 | Energy system transformation in energy supply sectors driven by a full phase-out of fossil fuels.

In addition, in line with the revisions made to Fig. 1 and Fig. 3, We also applied corresponding adjustments to the other figures (e.g., harmonising the colours for AIM, MESSAGEiX, and AR6, as well as for the scenarios).

3. Suggest adding the latest literature

Key achievements after 2024 have not been cited, such as The IEA's "clean energy transitions programme 2024" proposes the goal of "zero growth of fossil fuels by 2035", which is in line with the ZF2050 path and needs to be cited to enhance policy timeliness; Besides, most of the references for IAM's scenario research are from 10 years ago.

Response:

Thank you for the helpful comment. To improve policy timeliness, we have added the 2024 G7 and G20 summit statements, which highlight the post-COP28 momentum towards transitioning away from fossil fuels.

Page 2, line 40-43

Notably, in 2024, the Group of Seven (G7) agreed to phase out existing unabated coal power plants by the early 2030s, and the Group of Twenty (G20) leaders fully subscribed to the outcome of the COP28, in particular the UAE Consensus and the GST-1.

In addition, we updated the references to IAM and energy system model scenario research in the Introduction by adding more recent literature published after the late 2010s, as shown below.

- Page 2, line 49: Riahi et al., 2018; Guo et al., 2022; Oshiro & Fujimori, 2024
- Page 2, line 62: Lamb et al., 2024; Edelenbosch et al., 2024
- Page 2, line 64: Fuhrman et al., 2020
- Page 2, line 65: Schreyer et al., 2024
- Page 3, line 66: Speizer et al., 2024
- Page 3, line 70: Rose et al., 2022

4. There is room for further improvement in the practical value of the article

Firstly, if interdisciplinary perspectives are taken into account, the practical value of this article will be enhanced. At present, there is no cited research on "energy justice", which makes it difficult to provide theoretical support for "just transition"; In addition, ignoring behavioral economics literature, consumer behavior modeling was not included in the discussion. Besides, at present, the article does not propose an implementation roadmap and does not consider the dynamic possibilities of industrial economy and technological development.

Response:

Thank you for your suggestions to enhance the practical value of this paper.

First, as you pointed out, the section in the Discussion referring to "*transitioning away from fossil fuels in energy systems, in a just, orderly and equitable manner*" lacked sufficient theoretical support. To address this, we have added references and elaborated on the definition of just transition as well as the key aspects that should be emphasised in the context of the ZF scenarios.

Specifically, we have cited IPCC AR6 WG III Annex I¹ and Denton et al. (2022)² to illustrate what a just transition entails and to outline key priorities for ZF scenarios, including conventional 1.5 °C

scenarios, as shown below:

Page 18, line 521-525

In line with recent discussions on a just transition, which includes, for example, fairness in energy and climate decision-making processes and fostering of both international cooperation and coordinated multilateral actions⁵¹, the ZF scenarios should also emphasise mitigating the effects of energy transformations and envisioning an equitable decarbonised world⁵².

Furthermore, we have highlighted that, in the context of ZF scenarios, particular attention should be paid to countries that are highly dependent on fossil fuel revenues, citing Lazarus & van Asselt (2018)³ as follows:

Page 18, line 526-527

Notably, as the ZF scenarios require a more rapid and deeper phase-out of fossil fuels than typical 1.5°C scenarios, greater attention will need to be given to countries that are highly dependent on fossil fuel revenues and have limited capacity for transition⁵³.

Page 18, line 533-537

Although these findings are incomplete, they suggest significant impacts on current fossil fuel-exporting countries, highlighting the importance of carefully determining when and where to phase out fossil fuels beyond the uniform assumptions employed in this study. In addition, complementary policies supporting a just and equitable phase-out of fossil fuels will be essential to secure their cooperation⁵³.

Second, behavioural economics and consumer behaviour are important perspectives, but our study was designed to identify cost-optimal fossil fuel phase-out pathways, and therefore did not fully account for these aspects. To address this limitation, we have supplemented the discussion by examining potential barriers from a consumer behaviour perspective, using the transition from internal combustion engine vehicles to electric vehicles as an illustrative example, drawing on insights from previous study⁴.

Page 16, line 456-463

The potential for greater increases in energy investments and lifestyle changes is likely to pose challenges to the socioeconomic viability of the full phase-out of fossil fuels. For instance, the growing reliance on electricity and hydrogen in the transport sector requires a transition from internal combustion engine vehicles to battery and fuel cell electric vehicles, which may face barriers arising from consumer preferences and behavioural habits. Alternatively, fossil-fuel phase-out pathways with limited demand-side transformations based on synthetic fuels could be pursued, but these would involve higher costs³⁰.

Third, regarding the implementation roadmap, we acknowledge that the study does not explicitly propose a step-by-step transition plan or policy package, as our focus was on identifying cost-optimal global pathways toward full phase-out of fossil fuels. However, our modelling framework optimises the transition pathway over the period 2020–2100 rather than focusing on a single target year such as

2050. Therefore, the time-dependent results can serve as a foundation for developing more detailed implementation roadmaps.

Finally, with regard to the dynamic possibilities of industrial economy and technological development, our study partially considers such dynamics as exogenously given future trends. Specifically, the former is reflected through assumptions about future energy service demand, while the latter is represented by assumptions on cost reductions and efficiency improvements of technologies such as solar PV, wind, batteries and electrolyzers.

On the other hand, it is true that these dynamics are not endogenously modeled. To address this limitation and assess its implications, we conducted a supplementary analysis which is now included in this revision, incorporating macroeconomic demand responses through iterations between the MESSAGEix and MACRO⁵ models (Supplementary Fig. 8). This analysis showed that energy demand in the ZF scenarios decreased more significantly around mid-century compared to the Opt1.5C scenario, relative to the main analysis assuming inelastic demand (Supplementary Fig. 8a). However, the overall energy mix remained largely unchanged, despite incorporating such dynamic possibilities of the industrial economy and associated changes in energy demand (Supplementary Fig. 8b). This suggests that the impact of these dynamics on the overall energy transformations required for the full phase-out of fossil fuels within the context of this study’s main objective is likely to be limited. Similarly, even if the full phase-out of fossil fuels were to induce further technological development, its impact would likely remain confined to modest changes in the energy mix and system costs, without substantially affecting the qualitative discussions and conclusions of this study.

Supplementary Fig. 8 | Macroeconomic response of final energy demand in MESSAGEix. a, Percentage change in total final consumption in the ZF scenarios relative to the Opt1.5C scenario, for the MESSAGEix standalone and MESSAGEix-MACRO model. **b,** Final energy mix in 2050 and 2100 under the Opt1.5C, ZF2050, and ZF2100 scenarios in the MESSAGEix standalone and MESSAGEix-MACRO model. For details of the MESSAGEix-MACRO model, see Methods.

Reference:

1. IPCC. Annex I: Glossary [van Diemen, R., J.B.R. Matthews, V. Möller, J.S. Fuglestedt, V. Masson-Delmotte, C. Méndez, A. Reisinger, S. Semenov (eds)]. in *IPCC, 2022: Climate Change 2022: Mitigation of Climate Change. Contribution of Working Group III to the Sixth Assessment Report of the Intergovernmental Panel on Climate Change* (eds Shukla, P. R. et al.) (Cambridge University Press, Cambridge, UK and New York, NY, USA, 2022). doi:10.1017/9781009157926.020.
2. Denton, F. et al. Accelerating the transition in the context of sustainable development. *Climate Change 2022: Mitigation of Climate Change. Contribution of Working Group III to the Sixth Assessment Report of the Intergovernmental Panel on Climate Change* (2022) doi:10.1017/9781009157926.019.
3. Lazarus, M. & van Asselt, H. Fossil fuel supply and climate policy: exploring the road less taken. *Clim. Change* 150, 1–13 (2018).
4. Oshiro, K. et al. Alternative, but expensive, energy transition scenario featuring carbon capture and utilization can preserve existing energy demand technologies. *One Earth* 6, 872–883 (2023).
5. Messner, S. & Schrattenholzer, L. MESSAGE–MACRO: linking an energy supply model with a macroeconomic module and solving it iteratively. *Energy* 25, 267–282 (2000).

5. Lack of regional adaptability plan

This study focuses on global models without considering regional differences, and does not cross validate cross regional models with current models such as the Chinese MARKAL model and the European TIMES model. Suggest giving appropriate consideration to regional differentiation and proposing customized solutions for representative regions

Response:

Thank you for the valuable comment. We totally agree that the regional implications of fossil-fuel phase-out are highly important and of great interest. However, the primary objective of this study is to clarify the transformations required in the global energy system, as well as the associated challenges and opportunities, in order to achieve the full phase-out of fossil fuels. As already stated in the manuscript (Page 18, line 528-533 and Page 19, line 562-564), while a detailed national- and regional-level analysis lies beyond the scope of this paper, analysing potential heterogeneous regional pathways will be an important avenue for future research.

6. Some statements can be considered more rigorous and complete.

P2, L38-41. It is difficult to say the final decision language is “weaker” or not, because there must be not only scientific consideration but also geopolitical ones.

Response:

Thank you for your valuable comment. In response to your suggestion that the decision language cannot simply be described as weaker, we have revised the sentence to objectively state that a full phase-out is not explicitly mentioned.

Page 2, line 38-39

Although the final decision does not explicitly mention a full phase-out, momentum is growing for the phase-out of fossil fuels to become a key focus of climate policies moving forward from COP28.

P2, L42-43. It could be better to mention in which aspects or in which ways, the energy transformation pathways could be different.

Response:

Thank you for the helpful comment. While the subsequent paragraph already explains the reasons behind the differences in energy system transformation pathways between ZF and 1.5°C scenarios, we agree that indicating the relevant dimensions of difference earlier improves readability. We therefore inserted a brief sentence after the first sentence as follows.

Page 2, line 44-47

The decarbonisation and defossilisation of energy systems could follow different energy transformation pathways. These differences may concern the pace, sequence, and stringency of fossil fuel phase-out, especially in sectors that are technically and economically challenging to decarbonise.

Reviewer #2 (Remarks to the Author):

The manuscript is innovative and provides novel insights as to the role of COP28 outcomes as to the role of zero fossil [fuel] future energy systems and their differences as to more conventional net-zero carbon scenarios within the current body of literature.

This paper has the potential to become a seminal paper. It is incremental in methodological nature, but insightful and novel in the scenarios analyzed and the tone they potentially might set within the broader energy systems modelling and integrated assessment modelling community and more significantly, the policy bodies that consume these insights in their policy decision making process.

I did not find any Major flaws. Most of my comments are minor and suggestions in nature.

The paper is worthy of publication in my view. It could be published as is, but I expect my comments will aid in some clarity for the reader as minor revisions.

As always, some of my questions are answered by details, text, data and charts within the Extended Information section. However, one or two sentences should migrate forward into the main body. {GAS w/CCS and DACCS}

A point of note to the lead author. I believe you are an early stage career researcher, a PhD candidate that attended the IIASA YSSP - If this is the case, Excellent work - well done - keep it up.

The following comments are my main points for your consideration and minor revisions and are the same as per the attached excel file:

We sincerely appreciate your positive evaluation of our manuscript and your encouraging comments. We are grateful for your constructive suggestions, which have been very helpful in improving the clarity and overall quality of the paper. We have carefully addressed all your comments as detailed below.

ID Line Number Reviewer Comment

1 94 "Robust" - consider whether or not it is appropriate to use the term "Robust" to avoid confusion with "Robust Optimisation"

Response:

Thank you for your suggestion. To avoid the use of the word "robust" and to clarify our original intent, we have revised the sentence as follows.

Page 7, Line 95-100

To explore two distinct illustrative pathways for ZF energy systems and gain shared insights from the model ensemble, we employed two global energy system models: AIM-Technology (Asia–Pacific Integrated Model-Technology, hereinafter AIM)¹⁰, and MESSAGEix-GLOBIOM (Model for Energy Supply Strategy Alternatives and their General Environmental Impact combined with the Global Biosphere Management Model, hereinafter MESSAGEix)^{39–41}.

2 139-146 I'm wondering about the demand response options at this point, either through demand elasticities, energy service demand efficiencies or fuel switching. it seems a gap at this point

Response:

Thank you for the helpful comment. We would like to clarify how each demand-side response option is considered in this study:

- Demand elasticity: In the main analysis of this study, both AIM-Technology and MESSAGEix assume inelastic demand without considering any responsiveness of energy service demand to changes in energy prices. However, MESSAGEix can be coupled iteratively with MACRO, a macro-economic model, through which price-elastic demand response can be represented (see <https://docs.messageix.org/projects/models/en/latest/global/macro.html>). In the main analysis, we did not perform iterations with the MACRO model in order to maintain comparability with the AIM-Technology model. In this revision, we conducted a supplementary analysis by iteratively linking MESSAGEix with the MACRO model (Supplementary Fig. 8). The results showed that energy demand in the ZF scenario declined more sharply around mid-century compared to the Opt1.5C scenario, relative to the main analysis assuming inelastic demand (Supplementary Fig. 8a). Nevertheless, even when accounting for demand elasticity, the overall energy mix remained largely unchanged (Supplementary Fig. 8b).

Supplementary Fig. 8 | Macroeconomic response of final energy demand in MESSAGEix. a, Percentage change in total final consumption in the ZF scenarios relative to the Opt1.5C scenario, for the MESSAGEix standalone and MESSAGEix-MACRO model. **b,** Final energy mix in 2050 and 2100 under the Opt1.5C, ZF2050, and ZF2100 scenarios in the MESSAGEix standalone and MESSAGEix-MACRO model. For details of the MESSAGEix-MACRO model, see Methods.

- Energy service demand efficiency: In AIM-Technology, certain energy services are modeled with technologies that have different levels of efficiency, and the replacement with high-efficiency technologies is determined endogenously. In contrast, in MESSAGEix, energy efficiency improvements are statically modelled as exogenous trends over time and implicitly embedded within the demand projections.

- Fuel switching: Fuel switching is incorporated in both AIM-Technology and MESSAGEix. AIM-Technology represents energy service demands at a more disaggregated subsector level, allowing differences in the difficulty of fuel switching across subsectors to be captured more explicitly. For example, electrification technologies are available for passenger road transport demand, whereas aviation transport demand does not. In MESSAGEix, where energy service demands are more aggregated, such differences are captured through share constraints. For example, to reflect the presence of service demands that are difficult to electrify, an upper limit is placed on the share of electricity in the transport sector.

Taking the above into account, the differences between models in the shares of non-hydrocarbon fuels referred to here are likely to be predominantly driven by differences in the modeling of fuel switching. This is because energy service demand is inelastic, and efficiency improvements are either near the upper bounds assumed in the models or imposed exogenously, so they cannot explain the difference between models. Instead, differences in the scope of energy services that can be met by non-hydrocarbon fuels and in the availability of DAC-based synthetic hydrocarbon fuels are likely the main drivers.

First, in the Methods section, we made the following revisions to clarify how demand-side response options are assumed within each model.

Page 24, line 744-749

AIM-Technology assumes inelastic energy service demands. Certain energy services are modeled with technologies that have different levels of efficiency, and improvements in energy efficiency through the substitution by higher-efficiency technologies are determined endogenously. Fuel switching is represented by assuming different sets of available end-use technologies for each energy service at the subsector level.

Page 26, line 788-792

In this study, no iteration between MESSAGEix and MACRO was performed, and consequently inelastic useful energy demand was assumed. Energy efficiency improvements are modeled statically as exogenous trends over time and are implicitly embedded within the demand projections. The extent of fuel switching is represented by imposing share constraints.

Second, in the main text, we also added the following supplementary explanation.

Page 5, line 147-151

The main reason for this difference was the modelling of fuel switching on the end-use side, particularly in the scope of energy service demands that can be met by non-hydrocarbon technologies. In addition, MESSAGEix used in this study does not include direct air capture (DAC) and consequently does not consider DAC-based synthetic fuels, known as e-fuels, which further contributed to the difference.

3 154 In figure 1 - it's a great chart, information dense and spatially efficient - in panel f, I'm left wondering what GAS w/CCS is being used for in the Opt1.5C scenario. perhaps add a description to the caption

Response:

Thank you for your kind words about Fig. 1 and for your helpful suggestion. Most of the gas w/ CCS in the Opt1.5C scenario is used for power generation, with a smaller portion used for hydrogen generation. To clarify this point, we have added a brief clause to the existing sentence, specifying that natural gas power plants with CCS constitute the majority of fossil fuel with CCS as shown below.

Page 4, line 134-136

In the Opt1.5C scenario of MESSAGEix, natural gas power plants with CCS contribute as both a flexible generator and a bridging technology, constituting the majority of fossil fuel with CCS in this scenario.

4 209 Interesting finding with regard to Biomass mid century. Is there a trade off between land use availability and population dynamics in MESSAGE and AIM? does that impact on the supply curve later in the model horizon?

Response:

Thank you for your comment.

First, the AIM-Technology model itself does not internally account for the trade-off between land availability and population, however, the availability of bioenergy is constrained by externally estimated supply curves that take these factors into consideration. The supply potential of dedicated energy crops is classified into eight grades according to production costs. These supply potentials and costs are exogenously determined based on estimates¹ obtained from the AIM-Hub model², a recursive general equilibrium model, and the AIM-PLUM model³, a land use allocation model. Specifically, the AIM-Hub model is first used to estimate the future regional land demand for crops, afforestation, grassland, and forest, considering factors such as the trade-off between land-use availability and population dynamics. After this land demand is spatially downscaled to 0.5-degree grid cells using the AIM-PLUM model, the potential production of dedicated energy crops at a given price is calculated for the remaining land, and a corresponding supply curve is constructed. Other bioenergy supply potential including agriculture residue, forest residue, municipal solid waste and manure is obtained from GEA (2012)⁴.

Second, MESSAGEix scenario is based around SSP2 and thus considers the respective socio-economic developments, both for the energy and the AFOLU sectors. The incorporation of the AFOLU sector is modelled via an emulator^{5,6}, built on an ensemble of land-use pathways provided by GLOBIOM^{7,8}. Each of the pathways reflects population driven developments, both from the land-requirement and the nourishment perspective, each of which provides different levels of bioenergy supply at varying carbon-prices. The main sources of biomass in GLOBIOM are the forestry sector as well as dedicated short rotation tree plantations, which will have an interaction with area availability for the purpose of food production and/or pastures as well as land for other natural vegetation. Therefore, increased use of biomass can potentially result in trade-offs in areas dedicated for food

production purposes for example, while nevertheless maintaining the minimum per capita calorie intake criteria⁹. In the long-term, biomass supply curves consider population dynamics through explicitly modelling the living-space occupied, but also through managing different land-cover types in order to meet the demands associated with food supply and other products driven by population.

We have added the following description regarding the bioenergy supply potential in each model to the Methods section.

Page 24, line 735-739

The supply potential of dedicated energy crops is classified into eight grades according to production costs. These costs are exogenously determined based on Wu et al. (2019)⁵⁹. Other bioenergy supply potential including agriculture residue, forest residue, municipal solid waste and manure is obtained from GEA (2012)⁶⁰.

Page 25, line 776-780

To reduce computational costs, rather than running the full GLOBIOM model iteratively, the MESSAGEix model adopts a GLOBIOM emulator, built on an ensemble of land-use pathways provided by GLOBIOM. Each pathway reflects population-driven developments from both land-use and nutritional perspectives, resulting in different levels of bioenergy supply across a range of carbon prices.

References:

1. Wu, W. et al. Global advanced bioenergy potential under environmental protection policies and societal transformation measures. *GCB Bioenergy* 11, 1041–1055 (2019).
2. Fujimori, S. et al. SSP3: AIM implementation of Shared Socioeconomic Pathways. *Glob. Environ. Change* 42, 268–283 (2017).
3. Hasegawa, T., Fujimori, S., Ito, A., Takahashi, K. & Masui, T. Global land-use allocation model linked to an integrated assessment model. *Sci. Total Environ.* 580, (2016).
4. Global Energy Assessment Writing Team. *Global Energy Assessment: Toward a Sustainable Future*. (Cambridge University Press, Cambridge, 2012). doi:10.1017/CBO9780511793677.
5. Krey, V. et al. MESSAGEix-GLOBIOM Documentation. <https://pure.iiasa.ac.at/id/eprint/17115> (2020) doi:10.22022/iacc/03-2021.17115.
6. Fricko, O. et al. The marker quantification of the Shared Socioeconomic Pathway 2: A middle-of-the-road scenario for the 21st century. *Glob. Environ. Change* 42, 251–267 (2017).
7. Havlík, P. et al. Global land-use implications of first and second generation biofuel targets. *Energy Policy* 39, 5690–5702 (2011).
8. Havlík, P. et al. Climate change mitigation through livestock system transitions. *Proc. Natl. Acad. Sci.* 111, 3709–3714 (2014).
9. Frank, S. et al. Land-based climate change mitigation potentials within the agenda for sustainable development. *Environ. Res. Lett.* 16, 024006 (2021).

5 214 What are the end of horizon effects within the models of having this policy in the final period. would it impact on the robustness of signals to other sectors within the models (land use and climate?)

Response:

Thank you for the comment. The end-of-horizon effects are mainly related to the intertemporal optimisation in the MESSAGEix model, but we consider them to be small. In our study, the time horizon extends to 2110, even though the focus of the analysis is on the period up to 2100. This helps the model take decisions beyond 2100 and mitigates potential end-of-horizon effects, so the impact on signals to other sectors such as land use and climate is expected to be minimal.

We revised the description of the time horizon in the Methods.

Page 25 line 768-769

For time slices set at five-year intervals from 2025 to 2050 and ten-year intervals from 2055 to 2110, MESSAGEix optimises the total discounted system costs as the sum across these time slices.

6 219 It may be addressed latter - I'm wondering what are the costs and benefits of rapid scale up in energy crop production. land use impacts? biodiversity losses?

Response:

Thank you for the comment. Considering the implications for land use and biodiversity, we generally regard the rapid expansion of energy crop production as incurring costs rather than delivering benefits.

However, although the primary biomass supply in the ZF scenarios is larger and more rapid than in the Opt1.5C scenario, the associated impacts on land use and biodiversity are not expected to be particularly severe. For instance, Creutzig et al. (2015)¹ derived an upper limit of 100–300 EJ/yr for sustainable biomass use, while IPCC AR6 WGIII² identified thresholds of 100 EJ/yr for medium concern and 245 EJ/yr for high concern from a sustainability perspective. In our ZF scenarios, since biomass supply is constrained by sustainability limits within each model, it remains at most around 200 EJ/yr throughout the century, as shown in the figure included in the Supplementary Fig. 9. Nevertheless, as also noted in the main text, the implications for the land-use sector under the ZF scenarios warrant closer examination, and we highlight this as an important direction for future research.

We have added the figure shown below as Supplementary Fig. 9 and revised the corresponding part of the main text to refer to it.

Page 7 line 220-222

In particular, the trajectory of biomass shares indicates that the mid-century peak observed in the ZF scenarios can be avoided, implying that a rapid scale-up in energy crop production can also be avoided (Supplementary Fig. 9).

Supplementary Fig. 9 | Primary energy supply from biomass. Primary energy supply from biomass during the period 2020–2100. Creutzig et al. (2015)⁴ derived an upper limit of 100–300 EJ/yr for sustainable biomass use, while IPCC AR6 WGIII⁵ identified thresholds of 100 EJ/yr for medium concern and 245 EJ/yr for high concern from a sustainability perspective. Box plots illustrate primary energy supply from biomass in the IPCC AR6 C1 and C2 scenarios for 2030, 2050, 2070, and 2100. “n” denotes the number of available scenarios in each category.

References:

1. Creutzig, F. et al. Bioenergy and climate change mitigation: an assessment. *GCB Bioenergy* 7, 916–944 (2015).
2. IPCC. Scenarios and modelling methods. *Climate Change 2022: Mitigation of Climate Change. Contribution of Working Group III to the Sixth Assessment Report of the Intergovernmental Panel on Climate Change* (2022) doi:10.1017/9781009157926.022.

7 290 Figure 3 panel d - Very Interesting - the volatility in net-capacity increase raises obvious questions about the cashflow volatility within the technology supply chain, volatile labour markets and resulting price volatility for technologies - I wonder would showing net new capacity additions be more instructive and remove the retiring of end of life and fully depreciated technologies from the volatility. it might make it easier to understand this volatility

Response:

Thank you very much for your valuable comment. As you pointed out, capacity additions are a more meaningful indicator than net capacity increase, since they exclude technology retirements. We have therefore replaced the Fig. 3d with one based on capacity additions. In the manuscript we have replaced the term net capacity increase with capacity additions.

Page 9 line 275-278

A comparison of annual capacity additions for these technologies between the Opt1.5C and ZF scenarios revealed that the ZF scenarios exhibited more uneven growth rates and sharper peaks in

installation in the first half of the century compared to the Opt1.5C scenarios.

Fig. 3 | Energy system transformation in energy supply sectors driven by a full phase-out of fossil fuels. a, b, Power generation mixes and secondary energy flow in the Opt1.5C and ZF2050 scenarios of AIM-Technology (a) and MESSAGEix-GLOBIOM (b) in 2050. Bar plots on the left show power generation by energy source. Diagrams on the right show secondary energy flow associated with electricity and hydrogen-based energy carrier production. Grey shading represents energy losses including conversion, storage, and distribution and trade losses, curtailment, and electricity consumption for direct air capture (DAC). **c,** Total power and hydrogen generation in 2030, 2050, 2070, and 2100. Box plots illustrate power and hydrogen generation in the IPCC AR6 C1 and C2 scenarios for 2030, 2050, 2070, and 2100. **d,** Annual average capacity additions for solar and wind power, energy storage technologies, and electrolyser by decade in the Opt1.5C, ZF2100, and ZF2050 scenarios. “n” denotes the number of available scenarios in each category. For the regional results, please refer to Supplementary Fig. 10.

8 290 Following comment 7 - is there more or less volatility at the regional levels?

Response:

Thank you very much for your comment. The volatility that was partially smoothed out at the global level appears more prominently when examined at the regional levels as shown below. However, the qualitative feature described in the main text, that the peak in the first half of this century under the ZF scenarios is a common characteristic observed across different regions, remains unchanged.

We have now added the following figure to Supplementary Fig. 10. We have also included a reference to Supplementary Fig. 10 in the caption of Fig. 3 in the main text.

Supplementary Fig. 10 | Annual average capacity additions by decade. Annual average capacity additions for solar and wind power, energy storage technologies, and electrolyser by decade in Asia

(a), Latin America and the Caribbean (b), Middle East and Africa (c), OECD 90 and EU (d) and Reforming Economies (e) in the Opt1.5C, ZF2100, and ZF2050 scenarios.

Fig. 3 | Energy system transformation in energy supply sectors driven by a full phase-out of fossil fuels. a, b, Power generation mixes and secondary energy flow in the Opt1.5C and ZF2050 scenarios of AIM-Technology (a) and MESSAGEix-GLOBIOM (b) in 2050. Bar plots on the left show power generation by energy source. Diagrams on the right show secondary energy flow associated with electricity and hydrogen-based energy carrier production. Grey shading represents energy losses including conversion, storage, and distribution and trade losses, curtailment, and electricity consumption for direct air capture (DAC). c, Total power and hydrogen generation in 2030, 2050, 2070, and 2100. Box plots illustrate power and hydrogen generation in the IPCC AR6 C1 and C2 scenarios for 2030, 2050, 2070, and 2100. d, Annual average capacity additions for solar and wind power, energy storage technologies, and electrolyser by decade in the Opt1.5C, ZF2100, and ZF2050 scenarios. For the regional results, please refer to Supplementary Fig. 10.

9 305 Seems strange to talk about future scenarios in the past tense - I'm unsure of Journal guidelines but perhaps consider discussing scenarios in the continual present tense "when fossil fuel phase-out is achieved,..."

Response:

Thank you for your comment. We have revised the relevant part to use the present tense.

Page 11 line 309-311

When fossil fuel phase-out is achieved, CO₂ emissions from the energy sector are reduced in the ZF scenarios compared to the Opt1.5C scenarios by 2060–2070 (Fig. 4 and Supplementary Fig. 5a).

10 327 an annual carbon constraint is not a "carbon budget", it's a "carbon cap" and should be called such to avoid confusion

Response:

Thank you for the correction. We have changed "carbon budgets" to "carbon caps" in Line 319.

Page 11 line 330-332

By contrast, AIM, which applied annual carbon caps, did not exhibit such a rebound, as the predefined caps in each year constrained emissions regardless of earlier reductions.

11 321 The interaction between the two constraints, zero fossil and carbon budgets is interesting. In MESSAGE the ZF constraints binds before 2070 i expect and then the solution relaxes under a relatively larger carbon budget than a linear trajectory. I expect the marginal cost of carbon of those constraints are also displaying volatile non-linear behaviour. can you confirm that the discount rates and hurdle rates are static with time and do not change around 2070?

Response:

Thank you for your comment. We also interpret that in the ZF scenarios, the ZF constraints are binding up to around 2070, leading to deeper emission reductions than in the Opt1.5C scenario. After 2070, however, emissions become higher than in Opt1.5C under the more relaxed carbon budget. Nevertheless, as shown below, the marginal abatement costs do not exhibit particularly strong non-linear behaviour compared to the Opt1.5C scenario. In addition, we confirmed that the discount rate used in the MESSAGEix model, and it indeed remains static at 5% without any change around 2070.

Supplementary Fig. 11 | Carbon price in MESSAGEix scenarios.

12 321 Am i correct in remembering that AIM has dynamic recursive foresight and MESSAGE has perfect foresight? how does this and the annualised Carbon Cap (AIM) vs the Carbon Budget (MESSAGE) explain the post 2070 dynamics divergence in figure 4.a?

Response:

Thank you for the insightful comment. Yes, your understanding is correct: AIM adopts dynamic recursive foresight, while MESSAGE is based on perfect foresight.

In short, the reason why the difference between the annualized carbon cap and the carbon budget leads to the divergence in post-2070 dynamics lies in whether the carbon budget saved through deeper emission reductions in the first half of the century compared to the Opt1.5C scenario can be carried over into the latter half of the century. As you pointed out in Comment 11, the ZF constraint leads to deeper emission reductions in the first half of the century compared to the Opt1.5C scenario. In MESSAGE, this deeper emission reduction allows for more remaining carbon budget in the latter half

of the century. Consequently, net CO₂ emissions after 2070 are higher under the ZF scenario than under the Opt1.5C scenario. In contrast, AIM imposes a predefined annualized carbon cap. Thus, even if deeper emission reductions are achieved earlier, the model is still required to follow the same emissions trajectory as the Opt1.5C scenario after 2070. Accordingly, it is important to note that the cumulative emissions over the entire century are lower in AIM than in MESSAGE. To clarify this mechanism, we have added the following explanatory clauses.

Page 11 line 327-332

In MESSAGEix, which applied cumulative carbon budgets for the entire century, deeper emissions reductions in the first half of the century led to higher CO₂ emissions from the energy sector in the ZF scenarios than in the Opt1.5C scenario after 2070, as the saved carbon budget was carried over. By contrast, AIM, which applied annual carbon caps, did not exhibit such a rebound, as the predefined caps in each year constrained emissions regardless of earlier reductions.

13 377 I see Energy Demand (5.d) is skipped in this paragraph, but I'd welcome a sentence describing the demand sector investments within AIM and what they mean. a 10-20% increase in demand investments seems considerable and worthy of description.

Response:

Thank you for the helpful suggestion. While the result shown in Fig. 5d was already noted in page 13, line 376-377, we agree that further elaboration on its interpretation was needed. To address this, we extended the existing sentence, originally describing how the shift in end-use technologies may lead to lifestyle changes, to also reflect that this shift contributed to the observed increase in demand-side investments.

Page 13 line 383-387

In the ZF scenarios, non-hydrocarbon energy penetrated more deeply on the energy demand side compared to the Opt1.5C scenario (Fig. 5e). This result drove a shift from fossil fuel consumption technologies to electricity and hydrogen consumption technologies, which is associated with greater investments (Fig. 5d) and anticipated to result in lifestyle changes.

14 451 Is it worth mentioning the simplicity of the policy messaging around zero fossil fuel and how it might be easier to understand and implement for the general public?

Response:

Thank you for the suggestion. We agree that the simplicity of ZF scenarios can make them easier for the general public to understand and implement. Accordingly, we believe it is worth highlighting this point. We have therefore added the following sentence after the existing one that discusses the policy signal to fossil fuel producers.

Page 16 line 473-475

This simplicity may influence not only supply-side investment decisions but also facilitate understanding and implementation of ZF scenarios by the general public.

15 733 Within the document I believed that DAC is included as a technology in both models, but then in the EI it is stated that DAC is not in MESSAGE-IX. this should be clarified in the main body text.

Response:

Thank you for pointing this out. We have revised the main text to clarify whether MESSAGEix includes DAC in the following sentence, where this point could particularly influence the interpretation of the results.

Page 5 line 149-151

In addition, MESSAGEix used in this study does not include direct air capture (DAC) and consequently does not consider DAC-based synthetic fuels, which further contributed to the difference.

16 779 "We used the world V1.1 dataset" - for what? I don't understand this sentence

Response:

Thank you for your comment. The intention of this sentence is to clarify that the AR6 Scenario Explorer provides both the v1.0 and v1.1 versions of the dataset, and that we used the latter. To make this clear, we have added the following clarification to the sentence.

Page 27 line 840-844

We used the World v1.1 dataset (AR6_Scenarios_Database_World_v1.1), which was downloaded from the AR6 Scenario Explorer (<https://data.ece.iiasa.ac.at/ar6/>) on 28 October 2025.

17 868 ED Figure 1-Panelf -shows a stark result in the differences for Naturel Gas medum term outlook to 2060 in the high Opt1.5C case to considerably low Methane Gas in the ZF scenarios. Interesting - no action required. however this does link back to my previous question about what is GAS wCCS being used for in Opt1.5C? Industry?

Response:

Thank you for your comment. As noted in response to Comment No.3, in MESSAGEix's Opt1.5C scenario, the majority of natural gas with CCS is used in the power sector, with the remainder allocated to the hydrogen sector.

Reviewer #2 (Remarks on code availability):

I briefly reviewed the multiple Zenodo repositories and github repositories for the core MESSAGE-IX and AIM main branches.

It would take me some days/weeks to figure out if the code paper results are reproducible with the files provided.

The models are both open source and on github - this is excellent and following best practice in my view.

However, It is unclear to me if the files provided in the zenodo archives are simply model results files and post-processing scripts and not input parameters, or model constraints specific to this paper. (Carbon budgets, Carbon Caps and ZeroFossil constraint formulations)

I think it would be useful to archive the branch of the model version on github [& zenodo if desired] to make it easier to assess the replicability - i.e. keep all the model files needed to reproduce this paper in one location on github. This is common practice in some earth systems and data science communities, but not yet in IAM/ESOM communities.

Response:

Thank you for your suggestions. The set of files provided in the Zenodo archive at the time of the initial submission consisted of model output files and scripts for processing these outputs to generate figures. We have now clarified this point as follows in the Code availability section.

Page 28, line 879-880

The source code used for figure production from the scenario data is provided in the Zenodo repository (<https://zenodo.org/records/15254132>).

For the AIM-Technology model, the source of the model, such as the formulation, are available in a public GitHub repository. However, other parts, particularly the repositories containing licensed input data, are currently private repositories. Nevertheless, for the purposes including reproducibility verification during the review process, limited access to the input can be granted. We have clarified in the Code availability section as follows.

Page 28, line 880-883

The source code of the AIM-Technology model is available at the GitHub repository (https://github.com/KUAtmos/AIMTechnology_core). The input data of the AIM-Technology model is available from the authors upon reasonable request.

For MESSAGEix, in this revision we have archived an Excel file on Zenodo repository (<https://zenodo.org/records/17472924?token=eyJhbGciOiJIUzUxMiJ9.eyJpZCI6IjRiMzIyNzAyLWFmNTMtNDgzZi05NmExLWw5ZTY3NTE4ODE5MiIsImRhdGEiOnt9LCJyYW5kb20iOiJINmZkMmQxY2FjZmM1NWNhYWJiNTI1YVQwMTJkZmFIMyJ9.oW1C6MXSUwDQKEYnDA8yYZFenbCGzDkFIDKi33uWC41DVb-b2gLTbWs8J4LpSU-0zhpdkKTunqCrYldfO1I5bVw>) that contains all input data used in the scenarios. The link will only be available to the reviewers and editors during the review, and it will be made public after the paper is accepted. The scenarios of this study can be reproduced by loading the Excel file through the MESSAGEix command-line interface (CLI) as described in the documentation (<https://docs.messageix.org/projects/ixmp/en/latest/file-io.html>). We have added this statement to the Code availability as follows:

Page 28, line 885-887

The MESSAGEix scenario and model data, including all sets, parameters, variables, and equations,

have been deposited in the Zenodo repository (<https://zenodo.org/records/17472924>).

We appreciate the reviewers' comments and for the opportunity to submit a revised manuscript. The manuscript has been revised thoroughly according to both editorial and reviewer comments. This document contains point-by-point responses to each comment in blue. All changes in the revised manuscript are highlighted in yellow.

Reviewer #1 (Remarks to the Author):

Thanks for the revision of the manuscript and responses to previous comments. I still have some minor questions and comments:

General:

(1) The quantitative analysis of model results is quite detailed. However, more important conclusions need to be extracted, to further improve the practical value of the article. For example, with the solid quantitative analysis result, and all those recognized opportunities and challenges, is it possible to answer the question whether on earth the full phase-out of fossil fuels scenario is feasible or not?

Response:

Thank you for your insightful comment. While the manuscript already discusses the key characteristics of ZF energy systems, including their opportunities and challenges, from both quantitative and qualitative perspectives, we agree that the overall implications of the results can be stated more explicitly.

However, drawing a conclusion on the feasibility of a full phase-out of fossil fuels is not a question that can be answered based solely on global energy system models. Accordingly, the objective of this study is not to assess the feasibility of a full phase-out of fossil fuels, but rather to focus on the characteristics of the energy system transformation implied by such a phase-out together with the associated challenges and opportunities.

To better reflect this focus, we have added a key message at the beginning of the Discussion and Conclusions section to emphasise the main conclusion of the study prior to elaborated discussion, as shown below.

Page 15, line 435-439

Based on our results, a full phase-out of fossil fuels would require an energy system transformation that goes substantially beyond typical 1.5°C pathways, entailing non-negligible challenges and opportunities, thereby underscoring that defossilisation should not necessarily be equated with decarbonisation.

(2) Please explain about the reason/meaning of employing two IAM models in this study.

Response:

Thank you for your question. As described in Lines 95–100 of the manuscript, we employed two IAMs to explore two distinct illustrative pathways for zero-fossil energy systems and to gain shared insights from the model ensemble. This comparison also acts as a sophisticated sensitivity analysis framework that synthesizes the key signals from variations in techno-economic and technology diffusion parametrisation.

Previous studies, including Wilson et al. (2021) and Fujimori et al. (2025), have discussed that comparing results across multiple models, as commonly done in model intercomparison projects (MIPs), serves two main purposes. First, when model results agree on specific aspects, multi-model analyses can provide more robust insights by reducing biases arising from model-specific assumptions, such as model structure and parameter selections. Second, when model results show variation across models, such variation provides valuable information on the degree of uncertainty and allows a range of plausible future pathways to be depicted. Although the two-model ensemble used in this study is smaller in scale than typical MIPs, it adopts this underlying rationale by aiming both to extract insights that are robust across models and to explicitly acknowledge model-related uncertainty.

References:

1. Wilson, C., Guivarch, C., Kriegler, E. et al. Evaluating process-based integrated assessment models of climate change mitigation. *Climatic Change* 166, 3 (2021).
2. Fujimori, S. *et al.* Towards an open model intercomparison platform for integrated assessment models scenarios. *Nat. Clim. Change* **15**, 1156–1164 (2025).

Details:

(3) Line 40-43, please add the reference(s).

Response:

Thank you for your comment. We have added the relevant references to the G7 and G20 official agreement documents to support the statements in line 40-43. In addition, we have slightly revised the wording in the manuscript to better align with the language used in the original agreement texts, as follows.

Page 2, line 40-43

Notably, in 2024, the Group of Seven (G7) committed to phasing out existing unabated coal power generation during the first half of the 2030s⁴, and the Group of Twenty (G20) leaders fully subscribed to the outcome of COP28, in particular the UAE Consensus and the GST-1⁵.

References:

1. G7. G7 Climate, Energy and Environment Ministers' Joint Declaration. <https://www.g7italy.it/en/climate-energy-and-environment/> (2024).
2. G20. G20 Rio de Janeiro Leaders' Declaration. <https://www.gov.br/g20/en/documents/g20-rio-de-janeiro-leaders-declaration> (2024).

(4) Line 108-114, what is the difference between the Opt1.5C scenario developed in this study and the C1 and C2 scenarios included in AR6? The answer may solve the concern about the necessity of the Opt1.5C scenario.

Response:

Thank you for your comment. Based on the emission constraints considered in this study, there is no substantial difference between the Opt1.5C scenarios and the AR6 C1 and C2 scenarios. Nevertheless, we employed the Opt1.5C scenarios to examine the transformation of the energy system associated with a transition from a typical 1.5 °C pathway to a fossil phase-out scenario, while minimizing biases arising from model-specific assumptions.

In this study, the C1 and C2 scenarios are used to represent the broader solution space of typical 1.5 °C pathways and to provide a reference for comparison with the ZF scenarios, as described in line 118–120. By contrast, directly comparing the ZF scenarios with the C1 and C2 scenarios derived from different models or model versions would confound the effects of fossil phase-out due to inherent differences in model structure, assumptions, or versioning. To avoid this, we employ the Opt1.5C scenarios within the same model framework and assumptions as the ZF scenarios, with the fossil phase-out constraint as the only difference, thereby isolating the effects of fossil phase-out.

(5) Line 119-120, please check the data of text and figures. I read from the Fig. 1a that the proportion of fossil fuels are more than 40% and 60% in 2050 for the Opt1.5C scenario in AIM and MESSAGEix, respectively, which is different from 35% and 54% that mentioned in the text.

Response:

Thank you for carefully checking the consistency between the text and the figures. We confirmed that the fossil fuel shares reported in lines 119–120 (35% for AIM and 54% for MESSAGEix) are consistent with Fig. 1a when read along the fossil axis of the ternary diagram.

To avoid potential confusion in interpreting the ternary diagrams, we have clarified the reading of component shares in the caption of Fig. 1, as shown below.

Fig. 1 | Fossil fuel phase-out and energy system transformation. a–c, Energy mix for primary energy (a), power generation (b), and final energy (c) in 2050 shown as ternary diagrams. In these diagrams, the share of each component should be read from ticks parallel to the edge where that component equals zero.

(6) Fig 3d. under ZF scenarios, especially in the ZF2050 scenarios, what causes the fluctuations of these technologies?

Response:

Thank you for your question. The fluctuations in capacity additions under the ZF scenarios are primarily driven by rapid capacity expansion induced by the fossil fuel phase-out.

As discussed in the main text, the fossil phase-out requires a rapid scale-up of key technologies, such as solar and wind power, to support accelerated direct and indirect electrification. This leads to a pronounced peak in capacity additions in the first half of the century. Consequently, the cohorts of technologies installed during this initial peak reach the end of their technical lifetimes around similar periods, leading to a second peak in capacity additions to replace retiring capacity. This mechanism explains the multiple peaks in capacity additions observed in Fig. 3d.

In reality, technology retirements (planned or early retirement) would likely occur more gradually rather than at a fixed lifetime. However, the energy system model used in this study does not explicitly represent such variability in retirement timing, which can accentuate discrete peaks in capacity additions. Accounting for heterogeneous retirement timing would likely smooth these fluctuations to some extent. Importantly, the key qualitative insights, namely the presence of a major capacity expansion peak in the first half of the century and the overall increase in required capacity additions under fossil phase-out, are not sensitive to assumptions about retirement timing. Accordingly, the manuscript focuses on these features of energy system transformation driven by fossil fuel phase-out, as discussed in line 280–283.